# Activation of the cell wall integrity pathway negatively regulates TORC2-Ypk1/2 signaling through blocking eisosome disassembly in *Saccharomyces cerevisiae*
Wataru Nomura[1,2,3] ✉ & Yoshiharu Inoue [1] ✉

The target of rapamycin complex 2 (TORC2) signaling is associated with plasma membrane (PM) integrity. In *Saccharomyces cerevisiae*, TORC2-Ypk1/2 signaling controls sphingolipid biosynthesis, and Ypk1/2 phosphorylation by TORC2 under PM stress conditions is increased in a Slm1/2-dependent manner, under which Slm1 is known to be released from an eisosome, a furrow-like invagination PM structure. However, it remains unsolved how the activation machinery of TORC2-Ypk1/2 signaling is regulated. Here we show that edelfosine, a synthetic lysophospholipid analog, inhibits the activation of TORC2-Ypk1/2 signaling, and the cell wall integrity (CWI) pathway is involved in this inhibitory effect. The activation of CWI pathway blocked the eisosome disassembly promoted by PM stress and the release of Slm1 from eisosomes. Constitutive activation of TORC2-Ypk1/2 signaling exhibited increased sensitivity to cell wall stress. We propose that the CWI pathway negatively regulates the TORC2-Ypk1/2 signaling, which is involved in the regulatory mechanism to ensure the proper stress response to cell wall damage.

Cells maintain cellular homeostasis by sensing and responding to various extracellular and environmental conditions and adapting accordingly. Signal transduction pathways are crucial for transmitting extracellular stress signals to cells, and in most cases, the plasma membrane (PM) functions as a place to initiate and regulate these signaling pathways. Evolutionarily conserved protein kinase TOR (target of rapamycin) forms two distinct complexes, TOR complex 1 (TORC1) and TOR complex 2 (TORC2), and the signaling pathways to which these complexes contribute are essential for cell growth, metabolism, and stress response[1-4]. TORC2 activity is closely associated with the PM[5-7], and yeast TORC2 localizes at the domain just beneath the PM, which is known as the membrane compartment containing TORC2 (MCT) in the budding yeast *Saccharomyces cerevisiae*[6,8,9]. Additionally, a role of TORC2 in response to the PM stretch and stresses has been reported in neutrophils, *Dictyostelium*, and yeast, and TORC2 signaling is associated with the PM integrity and dynamics[6,10-13]. Therefore, TORC2 signaling is believed to play an important role in sensing the state of the PM[7,13,14].

In *S. cerevisiae*, TORC2 phosphorylates Ypk1 and its paralog Ypk2 (hereafter Ypk1/2), and Pkc1, all of which belong to the AGC (named initially for cAMP- and cGMP-dependent protein kinases and protein kinase C) kinase family, and TORC2 targets the conserved Ser/Thr residues within the turn and hydrophobic motifs of their C-terminal region[9,13,15,16]. TORC2 phosphorylates Ypk1/2, the ortholog of mammalian SGK1[17], at Ser644/641 in the turn motif involved in their stability and at Thr662/659 in the hydrophobic motif involved in their activity[9,13,15]. Ypk1/2 positively regulate sphingolipid biosynthesis, a primary lipid constituting eukaryotic PM, via increased phosphorylation of Orm1 and Orm2 (hereafter Orm1/2), which negatively regulate the first step enzyme of sphingolipid biosynthesis, serine palmitoyltransferase, and increased phosphorylation of Lag1/Lac1, which are catalytic subunits of the ceramide synthase[18-21]. Therefore TORC2-Ypk1/2 signaling plays a role in the biosynthesis of sphingolipids, which are essential for PM integrity[13,19,20,22,23]. Ypk1 phosphorylation at Thr662 in its hydrophobic motif by TORC2, which is necessary for Orm1/2 phosphorylation[19], is increased in response to the PM stretch triggered by

[1]Laboratory of Molecular Microbiology, Division of Applied Life Sciences, Graduate School of Agriculture, Kyoto University, Uji, Kyoto 611-0011, Japan. [2]Research Unit for Physiological Chemistry, the Center for the Promotion of Interdisciplinary Education and Research, Kyoto University, Kyoto 606-8501, Japan. [3]Present address: Division of Food Science and Biotechnology, Graduate School of Agriculture, Shinshu University, Nagano 399-4598, Japan.
✉e-mail: wnomura@shinshu-u.ac.jp; inoue.yoshiharu.5x@kyoto-u.ac.jp

hypo-osmotic shock and PM stress induced by the inhibition of sphingolipid biosynthesis[6,9,24]. It has been reported that PM-associated proteins, Slm1 and Slm2 (hereafter Slm1/2), which ensure the interaction between TORC2 and Ypk1/2, are crucial for the activation of TORC2-Ypk1/2 signaling caused by inhibitors of sphingolipid metabolism (e.g., myriocin and aureobasidin A, AbA) or PM stretch[6,9,24]. Slm1/2 possess an F-BAR domain and a pleckstrin homology (PH) domain, which can bind to membrane curvature and phosphatidylinositol 4,5-bisphosphate (PtdIns(4,5)$P_2$) enriched in PM, respectively[25,26]. Under nonstressed growth conditions, Slm1/2 localize prominently at furrow-like invagination structures of the PM organized by proteins, which are called eisosomes[6,27]. As a model for TORC2-Ypk1/2 signaling activation, it has been shown that when cells are exposed to hypo-osmotic shock or inhibitors of sphingolipid biosynthesis, Slm1/2 are released from the eisosomes and their localization to the MCTs are increased, enhancing Ypk1/2 phosphorylation by TORC2[6,9]. Additionally, it is known that eisosome disassembly is accelerated during the condition where the inhibitors of sphingolipid biosynthesis activate the TORC2-Ypk1/2 signaling[28–30]. Conversely, a reduced PM tension triggered by hyper-osmotic shock has been reported to emerge an invaginated PM structure in which PtdIns(4,5)$P_2$ accumulates (called the PtdIns(4,5)$P_2$-enriched structure; PES), and the TORC2-Ypk1/2 signaling is inactivated by sequestering TORC2 at the PES[12]. Therefore, PM dynamics are closely associated with an activity of TORC2-Ypk1/2 signaling, and eisosomes, Slm1/2, and PtdIns(4,5)$P_2$ are expected to play crucial roles in the mechanism of this association, although its regulatory mechanisms remain poorly understood.

Pkc1, another TORC2 target, is an ortholog of mammalian protein kinase Cs. TORC2-Pkc1 signaling is activated by methylglyoxal (MG), a natural glycolysis-derived metabolite, and modulates the Mpk1 mitogen-activated protein kinase (MAPK) cascade, which is a downstream signaling of Pkc1[16,31]. Pkc1 is also a component of the cell wall integrity (CWI) signaling pathway involved in the remodeling of cell walls and expression of stress response genes through Mpk1 MAPK, and the activation of the CWI pathway is induced by heat shock and cell wall stresses[31–34]. The stress-induced activation of the CWI pathway is mainly caused through the membrane proteins Wsc1 and Mid2, which are sensors for the CWI pathway[31,35], but these sensor proteins are dispensable for the MG-induced activation of TORC2-Pkc1 signaling[16], i.e., Pkc1 is thought to be modulated separately by the TORC2 and CWI pathways, respectively.

Recently, we have shown that MG also activates TORC2-Ypk1/2 signaling, and mutants of TORC2-Ypk1/2 signaling exhibit increased susceptibility to MG[36]. Meanwhile, we have also previously identified mutants defective in ergosterol biosynthetic process as MG-sensitive mutants in addition to mutants defective in the sphingolipid biosynthetic process[37]. In this study, we focused on the influence of ergosterol dynamics on TORC2-Ypk1/2 signaling, and found that edelfosine, a lysophosphatidylcholine analog, which has an affinity for sterols and affects the movement of PM sterols[38–40], inhibited the activation of TORC2-Ypk1/2 signaling. The inhibitory effect of edelfosine was exerted through CWI pathway activation, and eisosome structures are indispensable for this inhibitory effect. The activation of the CWI pathway blocked eisosome disassembly and the release of Slm1 from eisosomes, which were accelerated by the inhibition of sphingolipid biosynthesis. We also found that the inhibitory effect through CWI pathway activation on TORC2-Ypk1/2 signaling was involved in an adaptive response to cell wall stress. Our results provide insights into the mechanism by which the CWI pathway contributes to the regulation of TORC2-Ypk1/2 signaling.

## Results
### Edelfosine inhibits TORC2-Ypk1/2 signaling activation
TORC2-Ypk1/2 signaling positively regulates sphingolipid biosynthesis, and membrane stresses caused by sphingolipid metabolism inhibitors, e.g., myriocin and AbA, increase phosphorylation levels at the hydrophobic motif (Thr662/Thr659) of Ypk1/2 by TORC2[6,9,20]. Mutants of TORC2-Ypk1/2 signaling or sphingolipid biosynthesis exhibit increased

susceptibility to those inhibitors[19,41]. We have recently shown that MG enhances the phosphorylation levels of Ypk1/2, and mutants of TORC2-Ypk1/2 signaling exhibits increased susceptibility to MG[36]. A previous study using genetic screening showed that mutants defective in sphingolipid biosynthesis, csg2Δ and isc1Δ cells, exhibited MG sensitivity. Moreover, mutants defective in ergosterol biosynthesis, erg2Δ, erg3Δ, and erg6Δ cells, also had increased MG susceptibility[37] (Supplementary Fig. 1a). We anticipated the relationship between ergosterol and TORC2-Ypk1/2 signaling activation, but no severe defect in either AbA or MG-induced Ypk1/2 phosphorylation was observed in those mutants (Supplementary Fig. 1b). It has been suggested that the sterol-binding protein involved in the transport of ergosterol from the PM to the cortical ER is related to TORC2-Ypk1/2 signaling[42]. We therefore examined whether the ergosterol dynamics at the PM influence TORC2 signaling activation using sterol targeting reagents, amphotericin B, nystatin, filipin, and edelfosine[38,43]. As shown in Fig. 1a, b, amphotericin B, nystatin, and edelfosine strongly inhibited both AbA and MG-induced Ypk1/2 phosphorylation. Amphotericin B and nystatin are polyene antibiotics that are potent antifungal agents[43,44], and their toxicities were exerted under our experimental conditions (Fig. 1c). In contrast, edelfosine had no significant influence on cell viability (Fig. 1c), and therefore we focused the inhibitory effect of edelfosine on TORC2-Ypk1/2 signaling.

The potent antitumor lipid edelfosine, a lysophosphatidylcholine analog, targets cellular membranes and alters microdomain organization and morphology through its affinity for sterols[38–40]. The inhibitory effect of edelfosine on Ypk1/2 phosphorylation was also observed when cells were treated with myriocin in addition to AbA and MG (Fig. 1d). Although high edelfosine concentrations (20 μg/ml = 38.2 μM) have been reported to exert cytotoxicity and consequently inhibit the cell growth of yeast[39], AbA-induced Ypk1/2 phosphorylation was sufficiently abolished by treatment with 5 μM edelfosine that had little effect on cell growth (Fig. 1c–f). Since the activity of TORC2-Ypk1/2 signaling is necessary for AbA tolerance, we next examined the influence of edelfosine on AbA susceptibility. As shown in Fig. 1g, edelfosine-treated cells exhibited increased AbA susceptibility. A previous study on yeast screening for edelfosine-resisted mutants identified deletion mutants of retromer components (Vps29 and Vps35) involved in the retrograde transport of cargo proteins[45]. Neither the deletion of VPS29 nor VPS35 suppressed the inhibitory effect of edelfosine on Ypk1/2 phosphorylation (Supplementary Fig. 1c), suggesting that the impaired Ypk1/2 phosphorylation is not due to the cytotoxicity of edelfosine. Meanwhile, incorporation of edelfosine into cells is mediated by a flippase and is impaired by the mutation of Lem3, a flippase subunit[46]. LEM3 deletion suppressed the inhibitory effect of edelfosine on Ypk1/2 phosphorylation (Fig. 1h), indicating that edelfosine exerts its inhibitory effect on TORC2-Ypk1/2 signaling after incorporating into the cells. Besides sphingolipid biosynthesis inhibition, the mechanical stretch of PM provoked by hypo-osmotic shock induces transient activation of TORC2-Ypk1/2 signaling, which is more clear and persistent in mutant of the aquaglyceroporin Fps1[6,12]. As shown in Fig. 1i, edelfosine also inhibited the hypo-osmotic shock-induced Ypk1/2 phosphorylation in fps1Δ cells. Orm1 phosphorylation levels, a downstream effector of TORC2-Ypk1/2 signaling, are increased following treatment with myriocin or AbA, and these phosphorylations are observed as a mobility shift of Orm1 bands in SDS-PAGE followed by immunoblotting[19,21]. An increase in the band shift of Orm1 corresponding to the phosphorylated Orm1 following treatment with myriocin or AbA was diminished in the presence of edelfosine (Fig. 1j). These data indicate that edelfosine inhibits TORC2-Ypk1/2 signaling activation.

### Edelfosine activates the CWI pathway
In addition to Ypk1/2, Pkc1, a component of the CWI pathway, has also been identified as a direct substrate of TORC2, and TORC2 regulates Pkc1 through its phosphorylation of the hydrophobic motif (Ser1143), which is independent of the CWI pathway[16,31] (Fig. 2a). To determine whether edelfosine inhibits TORC2-Pkc1 signaling, the levels of Pkc1 phosphorylation at

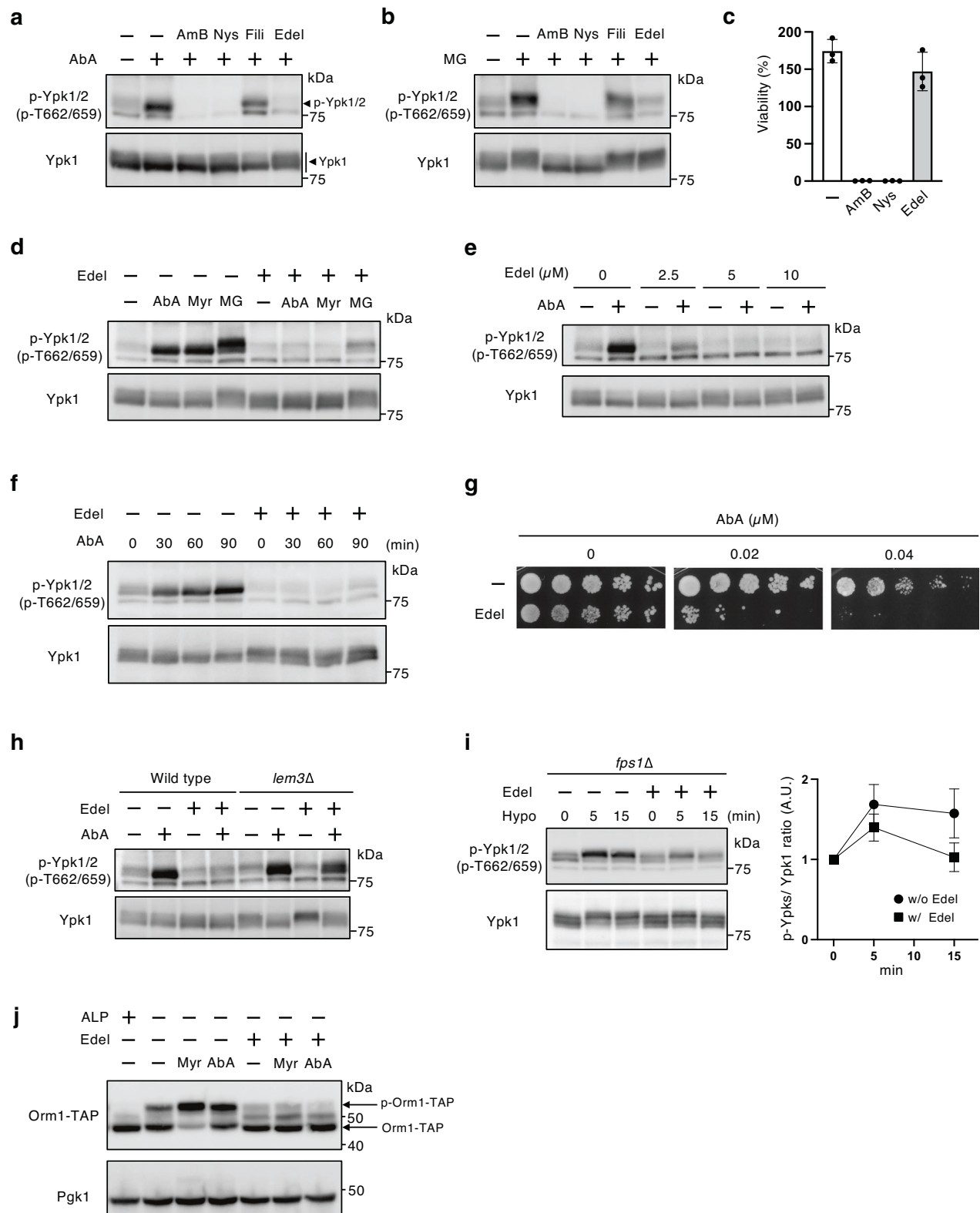

Ser1143 were checked. Pkc1 phosphorylation levels in the steady state were highly kept, and edelfosine had no strong influence on it (Fig. 2b). Although edelfosine did not inhibit TORC2-Pkc1 signaling, we noticed that the phosphorylation levels of Mpk1 MAP kinase, a downstream effector of the CWI pathway, were enhanced following treatment with edelfosine in a Pkc1-dependent manner (Figs. 2b and 2c). Additionally, the edelfosine-induced

Mpk1 phosphorylation was diminished by the deletion of *WSC1* and *MID2*, which are major sensors of the CWI pathway localized at PM[31,35] (Fig. 2d). These findings suggest that edelfosine does not inhibit the TORC2-Pkc1 signaling, but it does cause the CWI pathway activation in a Pkc1-dependent manner. The CWI pathway activation is induced by heat shock and cell wall stresses, such as treatments with cell wall-perturbing agents[32-34]. Calcofluor

**Fig. 1 | Edelfosine inhibits TORC2-Ypk1/2 signaling. a**, **b** Wild-type (BY4741) cells were cultured in SD medium until the log phase of growth and pretreated with 5 μM amphotericin B (AmB), 5 μM nystatin (Nys), 5 μg/ml filipin (Fili), or 5 μM edelfosine (Edel) for 15 min, and were then further grown for another 60 min after the addition of 1.25 μM AbA (**a**) or 15 mM MG (**b**). Phosphorylated Ypk1/2 or Ypk1 was detected with anti-phosphorylated Ypk1T662 and Ypk2T659, or anti-Ypk1 anti-body, respectively; a band of phosphorylated Ypk1/2 or Ypk1 was indicated by arrowhead. **c** Wild-type (BY4741) cells were cultured in SD medium until the log phase of growth and treated with 5 μM amphotericin B (AmB), 5 μM nystatin (Nys), or 5 μM edelfosine (Edel) for 60 min, and viability was determined by counting the number of colonies after plating cells on YPD agar plates. The viability before the addition of the reagents was set as 100%. Data are from three independent experiments (mean ± s.d.). **d** Wild-type (BY4741) cells were cultured in SD medium until the log phase of growth and pretreated with 5 μM edelfosine for 15 min, and were then further grown for another 60 min after addition of 1.25 μM AbA, 1.25 μM myriocin (Myr), or 15 mM MG. The phosphorylated Ypk1/2 or Ypk1 was detected as described in (**a**). **e** Wild-type (BY4741) cells were cultured in SD medium until the log phase of growth and pretreated with 0, 2.5, 5, or 10 μM edelfosine for 15 min, and further grown for another 60 min after addition of 1.25 μM AbA. Phosphorylated Ypk1/2 or Ypk1 was detected as described in (**a**). **f** Wild-type (BY4741) cells were cultured in SD medium until the log phase of growth and pretreated with 5 μM

edelfosine for 15 min, and treated with 1.25 μM AbA for the prescribed times. Phosphorylated Ypk1/2 or Ypk1 was detected as described in (**a**). **g** Wild-type (BY4741) cells were cultured in SD medium until the log phase of growth and treated using 5 μM edelfosine for 60 min, and then were spotted in 5-fold serial dilution onto SD agar plates with AbA. The plates were scanned after incubation for 3 days at 28 °C. **h** Wild type (BY4741) and *lem3Δ* cells were cultured in SD medium until the log phase of growth and pretreated with 5 μM edelfosine for 15 min and were then further grown for another 60 min after the addition of 1.25 μM AbA. Phosphorylated Ypk1/2 or Ypk1 was detected as described in (**a**). **i** *fps1Δ* cells were cultured in SD medium with 1 M sorbitol until the log phase of growth and pretreated with 5 μM edelfosine for 45 min. Cells were collected and transferred to SD medium without sorbitol to expose them to hypo-osmotic shock. Phosphorylated Ypk1/2 or Ypk1 at the prescribed times after hypo-osmotic shock was detected as described in (**a**). The intensity of the immunoreactive bands was quantified by image analysis. The ratio of phosphorylated Ypk1/2 (p-Ypks) to that of total Ypk1 was normalized to that of the sample at 0 min set as one. Data are from three independent experiments (mean ± s.d.). **j** *ORM1-TAP* cells were cultured in SD medium until the log phase of growth and pretreated with 5 μM edelfosine for 15 min, and were then further grown for another 120 min after the addition of 1.25 μM AbA or 1.25 μM myriocin. A portion of the untreated control sample was treated with calf intestinal alkaline phosphatase. Orm1-TAP was detected using PAP antibody.

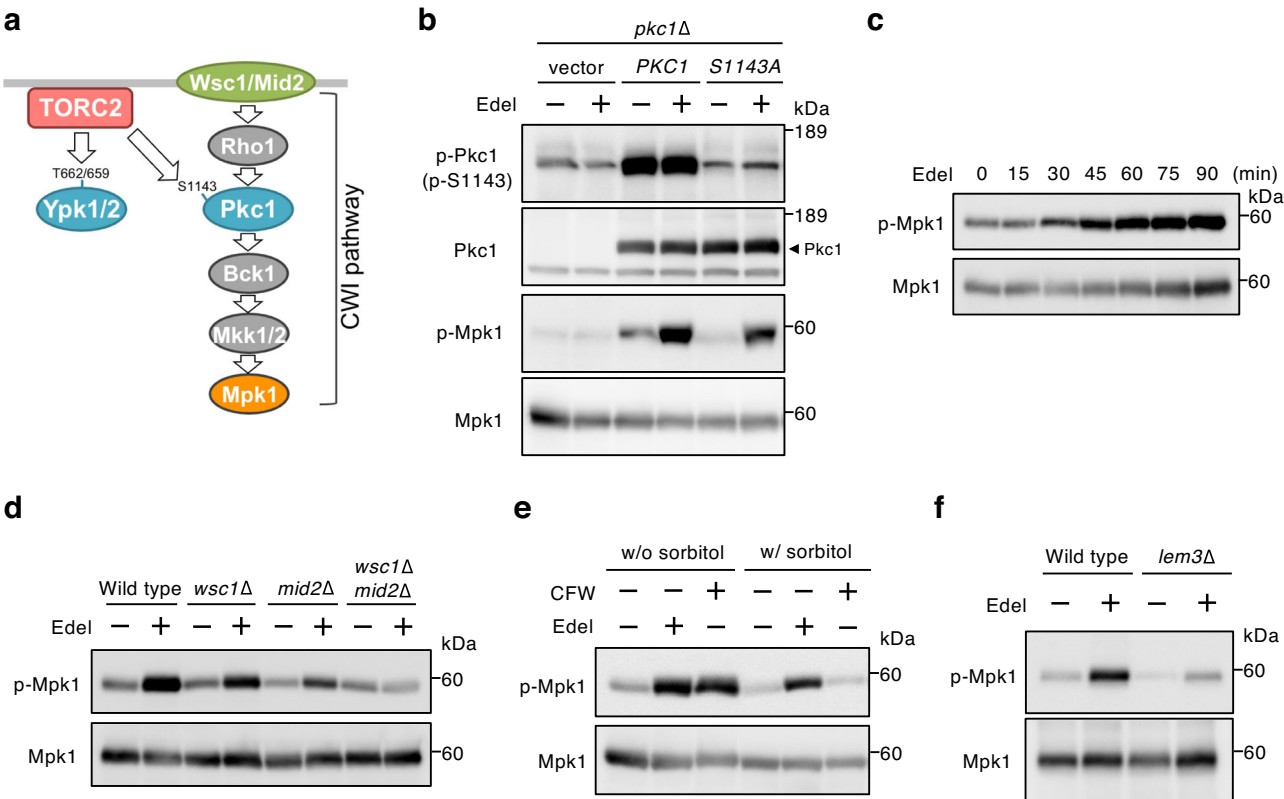

**Fig. 2 | Edelfosine activates the CWI pathway. a** Conserved sites of Pkc1 in addition to Ypk1/2 are phosphorylated by TORC2. Although Pkc1 is a component of the CWI pathway, TORC2 regulates Pkc1 independently of the CWI pathway. **b** *pkc1Δ* (DL376)[79] cells carrying an empty vector, YEp352-*PKC1* (pFR22), or YEp352-*PKC1S1143A* (pFR74)[80] were cultured in SD medium with 1 M sorbitol until the log phase of growth and treated with 5 μM edelfosine for 60 min. Phosphorylated Pkc1, Pkc1, phosphorylated Mpk1, or Mpk1 was detected with anti-phosphorylated Pkc1S1143, anti-Pkc1, anti-phosphorylated Mpk1, or anti-Mpk1 antibody, respectively; a band of Pkc1 is indicated by an arrowhead. **c** Wild-type (BY4741) cells were cultured in SD medium until the log phase of growth and treated with 5 μM

edelfosine for the prescribed times. Phosphorylated Mpk1, or Mpk1 was detected as described in (**b**). **d** Wild-type (BY4741), *wsc1Δ*, *mid2Δ*, and *wsc1Δmid2Δ* cells were cultured in SD medium until the log phase of growth and treated with 5 μM edel-fosine for 60 min. The phosphorylated Mpk1 or Mpk1 was detected as described in (**b**). **e** Wild-type (BY4741) cells were cultured in SD medium with or without 1 M sorbitol until the log phase of growth and treated with 5 μM edelfosine or 40 μg/ml CFW for 60 min. Phosphorylated Mpk1 or Mpk1 was detected as described in (**b**). **f** Wild-type (BY4741) and *lem3Δ* cells were cultured in SD medium until the log phase of growth, and the phosphorylation of Mpk1 induced by edelfosine was determined as described in (**b**).

white (CFW) that binds to chitin in the yeast cell wall and interferes with normal wall assembly induces the CWI pathway activation through the membrane sensors, and consequently leads to an increase in Mpk1 phosphorylation[33]. The increased Mpk1 phosphorylation by treatment with

cell wall-perturbing agents, including CFW, is attenuated by 1 M sorbitol for osmotic support[33]. The CFW-induced Mpk1 phosphorylation was abolished by culturing cells in 1 M sorbitol; however, it was not observed in the case of edelfosine treatment (Fig. 2e). Meanwhile, the edelfosine-induced Mpk1

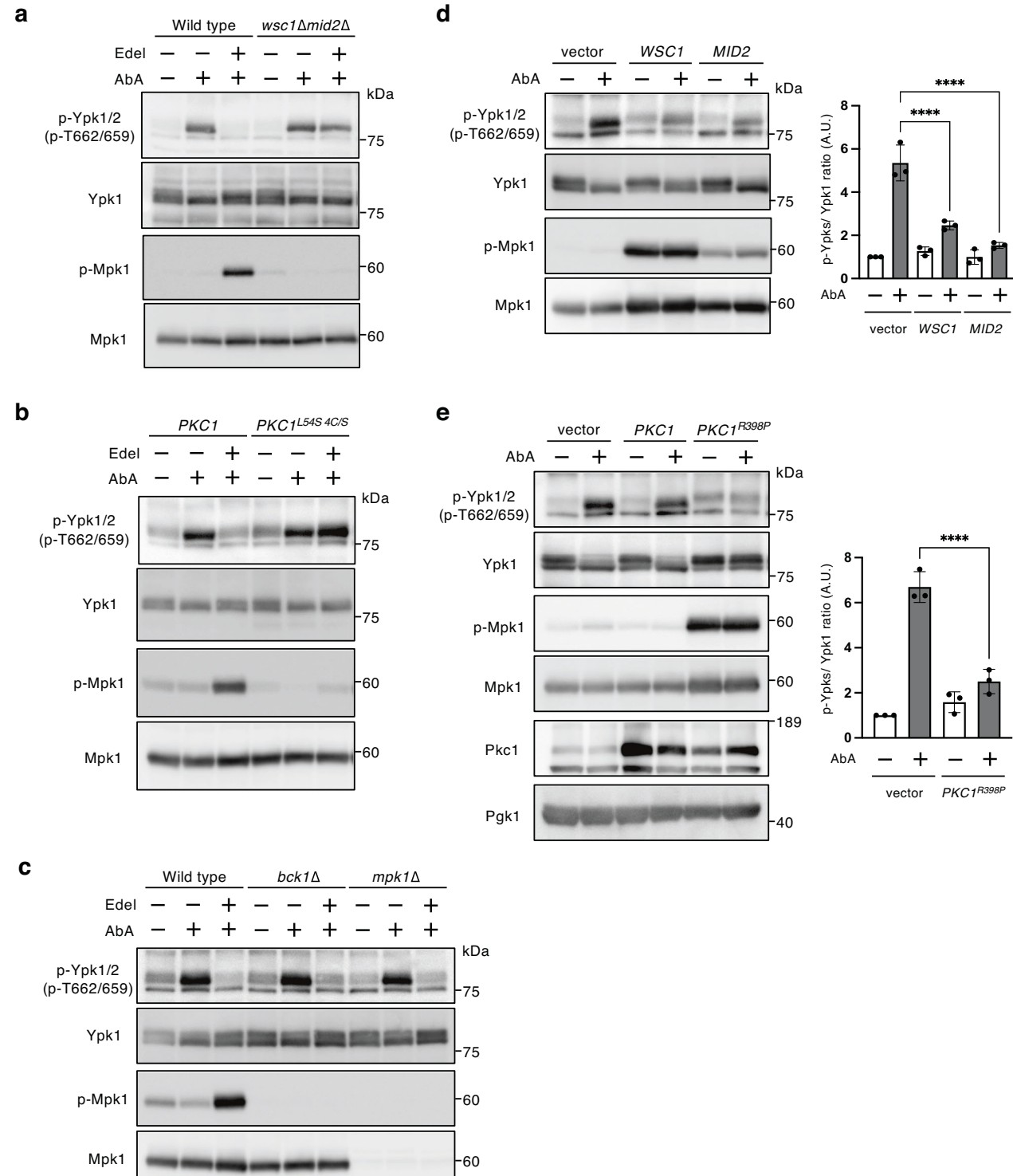

Communications Biology | (2024)7:722

phosphorylation was diminished in *lem3Δ* cells (Fig. 2f). These results suggest that edelfosine activates the CWI pathway by incorporating into PM rather than causing cell wall damage.

**Activation of the CWI pathway inhibits TORC2-Ypk1/2 signaling**

To determine whether the edelfosine-induced activation of the CWI pathway is involved in TORC2-Ypk1/2 signaling inhibition, the inhibitory effect of edelfosine in mutants defective in the CWI pathway was examined. As shown in Fig. 3a, the inhibitory effect of edelfosine on Ypk1/2 phosphorylation was attenuated in *wsc1Δmid2Δ* cells. In the CWI pathway, Pkc1 is

activated by physical interaction with a GTP-bound form of small GTPase Rho1 via its HR1 (homology region 1) and C1 (conserved region 1) domains[31,47–49]. Mutations of both the HR1 and C1 domains in order to interfere with the physical interaction between Rho1 and Pkc1, i.e., substitution of Leu54 with Ser in HR1 and four Cys residues with Ser in the C1 domain[48,50], abolished the edelfosine-induced Mpk1 phosphorylation and attenuated the inhibitory effect of edelfosine on Ypk1/2 phosphorylation (Fig. 3b). Conversely, no marked attenuation of the inhibitory effect of edelfosine was observed in mutants of the Mpk1 MAP kinase cascade components, Bck1 and Mpk1, which are downstream Pkc1 effectors

**Fig. 3 | CWI pathway inhibits TORC2-Ypk1/2 signaling. a** Wild-type (BY4741) and *wsc1Δmid2Δ* cells were cultured in SD medium until the log phase of growth and pretreated with 5 μM edelfosine for 15 min, and were further grown for another 60 min after the addition of 1.25 μM AbA. Phosphorylated Ypk1/2, Ypk1, phosphorylated Mpk1, or Mpk1 was detected with anti-phosphorylated Ypk1[T662] and Ypk2[T659], anti-Ypk1, anti-phosphorylated Mpk1, or anti-Mpk1 antibody, respectively. **b** *PKC1* and *PKC1[L54S_4C/S]* cells were cultured in SD medium with 1 M sorbitol until the log phase of growth and pretreated with 5 μM edelfosine for 15 min, and were then further grown for another 60 min after addition of 1.25 μM AbA. Phosphorylated Ypk1/2, Ypk1, phosphorylated Mpk1, or Mpk1 was detected as described in (**a**). **c** Wild-type (BY4741), *bck1Δ*, and *mpk1Δ* cells were cultured in SD medium until the log phase of growth and pretreated with 5 μM edelfosine for 15 min, and were then further grown for another 60 min after the addition of 1.25 μM AbA. Phosphorylated Ypk1/2, Ypk1, phosphorylated Mpk1, or Mpk1 was detected as described in (**a**). **d** Wild-type (BY4741) cells carrying an empty vector, YCpLG-*WSC1*, or YCpLG-*MID2* were cultured in raffinose medium until the log phase of growth, and galactose was added to the medium to a concentration of 2%. After incubation for 2 h, cells were treated with 1.25 μM AbA for 60 min. Phosphorylated Ypk1/2, Ypk1, phosphorylated Mpk1, or Mpk1 was detected as described in (**a**). The intensity of the immunoreactive bands was quantified by image analysis. The ratio of phosphorylated Ypk1/2 (p-Ypks) to that of total Ypk1 was normalized to that of the control sample with the empty vector set as one. Data are from three independent experiments (mean ± s.d.). **e** Wild-type (BY4741) cells carrying an empty vector, YCpLG-*PKC1*, or YCpLG-*PKC1[R398P]* [81,82] were cultured and treated with AbA as described in (**d**). The ratio of phosphorylated Ypk1/2 was determined as described in (**d**). Data are from three independent experiments (mean ± s.d.). ****$p < 0.0001$ (one-way ANOVA with Tukey's multiple comparisons test).

(Fig. 3c). These results suggest that the CWI pathway activation induced by edelfosine inhibits TORC2-Ypk1/2 signaling in a Mpk1 MAP kinase cascade-independent manner.

Next, we examined whether the CWI pathway activation is sufficient to inhibit TORC2-Ypk1/2 signaling. An overexpression of *WSC1* or *MID2* activated the CWI pathway and then increased the Mpk1 phosphorylation levels, under which conditions the activation of TORC2-Ypk1/2 signaling was significantly repressed (Fig. 3d). Additionally, an overexpression of a GTP-locked Rho1 mutant (Rho1[Q68L]) or a constitutively active Pkc1 mutant (Pkc1[R398P]) also attenuated the AbA-induced Ypk1/2 phosphorylation (Fig. 3e, and Supplementary Fig. 2), suggesting that Pkc1 activation through the CWI pathway causes the inhibitory effect on TORC2-Ypk1/2 signaling.

### Effect of activation of the CWI pathway on eisosome disassembly and Slm1 release from eisosomes

F-BAR- and PH-domain-containing proteins Slm1 and Slm2 play major roles in TORC2-Ypk1/2 signaling activation, and especially in the recruitment of Ypk1 to PM, including MCTs[6,9,22,51] (Fig. 4a). Under normal conditions, Slm1/2 preferentially localize at eisosomes, whereas the inhibition of sphingolipid biosynthesis reduces the ratio of their localization at eisosomes and increases their localization at MCTs[6,9]. Meanwhile, sphingolipid biosynthesis inhibition also promotes the disassembly of eisosome structures[28–30]. Although it is unclear whether the dynamics of eisosome structures are required for the activation of TORC2-Ypk1/2 signaling, eisosome disassembly occurs simultaneously with the delocalization of Slm1/2 from eisosomes under inhibitory conditions of sphingolipid biosynthesis (Fig. 4a). A previous study of phosphoproteomic analysis has indicated that overexpression of an active *PKC1* mutant influences the phosphorylation status of some eisosome components, including a core components Pil1 and Lsp1, and the phosphorylation of Pil1 is involved in eisosome assembly[52]. These observations suggest the possibility that eisosomes participate in the inhibitory effect through the CWI pathway on TORC2-Ypk1/2 signaling. To confirm this possibility, we examined whether edelfosine affects the eisosome using microscopy with mCherry-tagged Sur7, a transmembrane component of eisosome[27], as an eisosome marker. Under normal conditions, Sur7-mCherry patches at the PM were clearly observed as previously reported[27,53] (Fig. 4b). To determine the degree of eisosome assembly, the relative fluorescence of Sur7-mCherry was measured along the PM. The clarity of patchy localization of Sur7-mCherry, indicated by peaks, was decreased, and its diffusion throughout the PM was enhanced following treatment with AbA, i.e., eisosome disassembly was enhanced. AbA-induced diffusion of Sur7-mCherry was blocked in the presence of edelfosine (Fig. 4b), and furthermore the blockage of Sur7-mCherry diffusion was also verified by the overexpression of *WSC1* or *MID2* (Fig. 4c). The effect of edelfosine on eisosome disassembly was also verified using Lsp1-mCherry (Supplementary Fig. 3a). Meanwhile, TORC2 localization, which forms a punctate pattern beneath the PM, was not markedly altered following treatment with edelfosine (Supplementary Fig. 3b). These results suggest that the CWI pathway blocks eisosome disassembly induced by the inhibition of sphingolipid biosynthesis.

Based on findings that Slm1/2 are released from eisosomes following sphingolipid biosynthesis inhibition[6,24], we hypothesized that the CWI pathway activation also blocked the release of Slm1/2 from eisosomes, which is required for Ypk1/2 phosphorylation by TORC2, in addition to eisosome disassembly. To test this hypothesis, we next examined whether edelfosine blocks the release of Slm1 from eisosomes following treatment with AbA. Slm1-GFP and Sur7-mCherry were mainly observed as patches at PM, most of which were colocalized, and an overlap of the fluorescence signals was detected (Fig. 4d). Treatment with AbA decreased and diffused Sur7-mCherry patches, resulting in less overlap between Slm1-GFP and Sur7-mCherry patches, and this AbA-induced reduction of overlap was suppressed by edelfosine (Fig. 4d). The reduction in the colocalization of Slm1-GFP and Sur7-mCherry indicating Slm1 release from eisosomes following treatment with AbA was also observed in the cortical view, and edelfosine suppressed the AbA-induced reduction in the colocalization (Fig. 4e). Additionally, Slm1-GFP and Sur7-mCherry colocalization was enhanced under the presence of edelfosine (Fig. 4e). These edelfosine effects on Slm1-GFP and Sur7-mCherry colocalization were also observed in the case of the overexpression of *PKC1[R398P]*, a constitutively active mutant of Pkc1 (Fig. 4f). These data suggested that the CWI pathway activation simultaneously blocked eisosome disassembly and Slm1 release from eisosomes, which were caused by sphingolipid biosynthesis inhibition.

### Eisosome structures are necessary for the inhibitory effect of the CWI pathway on TORC2-Ypk1/2 signaling

If blockage of eisosome disassembly and Slm1 release from eisosomes are responsible for the inhibitory effect on TORC2-Ypk1/2 signaling, the inhibitory effect would be exerted via eisosomes. We then examined whether the inhibitory effect through the CWI pathway on TORC2-Ypk1/2 signaling depends on eisosome structures using a *pil1Δlsp1Δ* mutant, in which typical eisosomes disappear and instead a few large clusters, called eisosome remnants, emerge[27]. As shown in Fig. 5a, although the increased Ypk1/2 phosphorylation levels induced by AbA in *pil1Δlsp1Δ* cells were comparable with those in wild-type cells, the inhibitory effect of edelfosine on TORC2-Ypk1/2 signaling activation was markedly attenuated in *pil1Δlsp1Δ* cells. The inhibitory effect of *MID2* overexpression was also attenuated in *pil1Δlsp1Δ* cells (Fig. 5b). Additionally, treatment with edelfosine increased AbA susceptibility (Fig. 1g), which was suppressed by deletion of *PIL1* and *LSP1* (Fig. 5c). These results suggest that eisosome structures are necessary to exert the inhibitory effects through the CWI pathway on TORC2-Ypk1/2 signaling.

Pil1 is known to be phosphorylated at multiple sites[28,29,52], in which phosphorylations at Ser230 and Thr233 are induced by Pkc1 signaling and contribute to eisosome assembly[52]. Pkc1 does not directly phosphorylates at Ser230 and Thr233 of Pil1, but is involved in these phosphorylations through Mpk1[52]. We determine whether Pil1 phosphorylations at Ser230 and Thr233 are involved in the inhibitory effect on TORC2-Ypk1/2 signaling activation. The inhibitory effect of edelfosine was also sufficiently attenuated by *PIL1* deletion, and this attenuation of the inhibitory effect in *pil1Δ* cells was reverted by introducing wild-type *PIL1* and *PIL1[S230A/T233A]*,

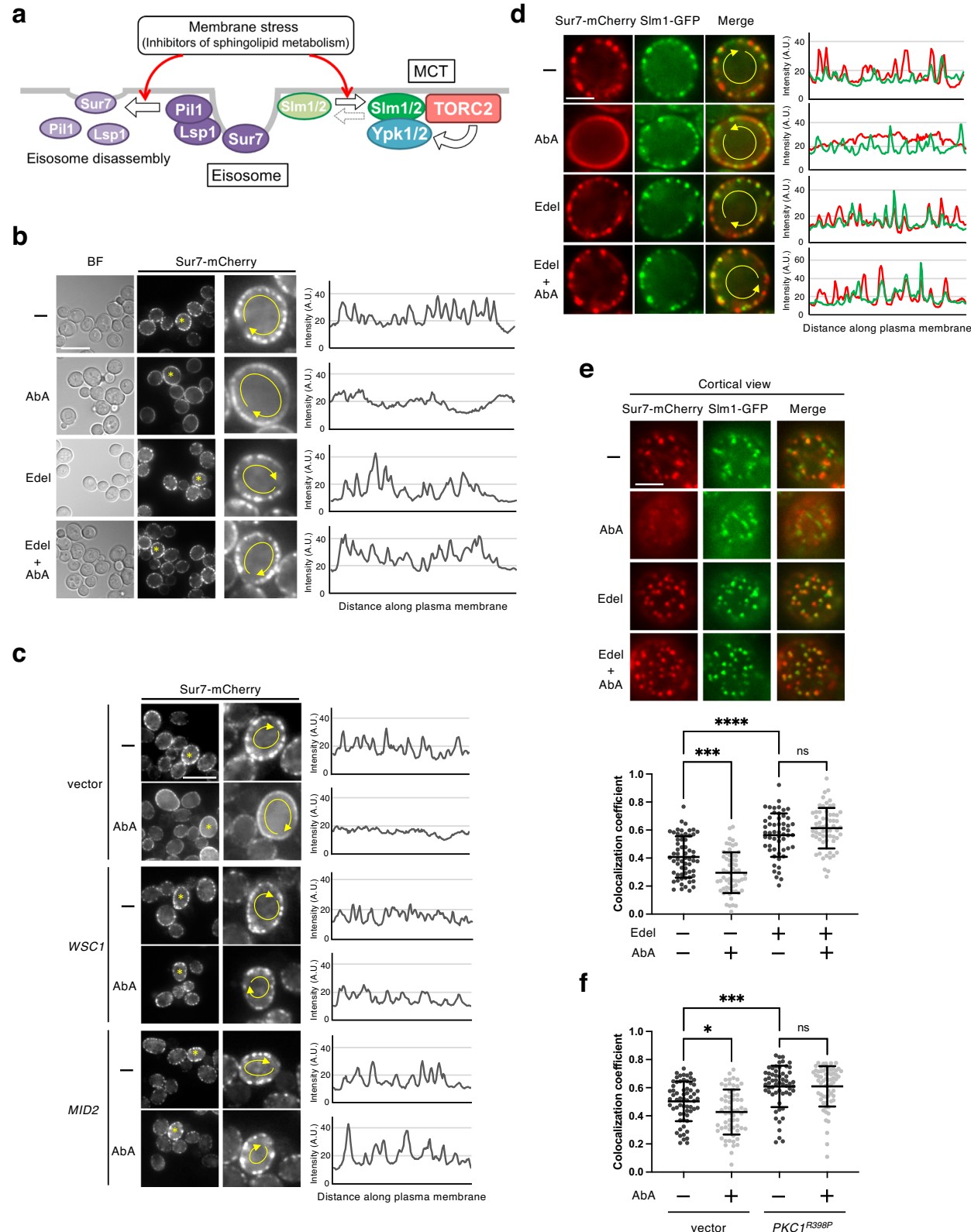

a mutant of Ala substitutions of Ser230 and Thr233 (Fig. 5d). This result indicated that the phosphorylation status of Pil1 at Ser230 and Thr233, which are targeted by Mpk1, is not involved in the inhibitory effect through the CWI pathway on TORC2-Ypk1/2 signaling, consistent with the result that the inhibitory effect of edelfosine was independent of the Mpk1 MAP kinase cascade (Fig. 3c).

**The inhibitory effect through the CWI pathway on TORC2-Ypk1/2 signaling contributes to the cell wall stress response**

The CWI pathway is activated in response to cell wall damage in order to maintain cell wall integrity. To determine whether the inhibitory effect through the CWI pathway on TORC2-Ypk1/2 signaling is exerted under cell wall-damaged conditions, the effect of the cell wall-perturbing agent

**Fig. 4 | CWI pathway blocks eisosome disassembly and Slm1 release from eisosome. a** PM stress induces eisosome disassembly and the release of Slm1/2, which are eisosome-localized proteins necessary for activating TORC2-Ypk1/2 signaling, from eisosome. **b** *SUR7-mCherry* cells were cultured in SD medium until the log phase of growth and pretreated with 5 µM edelfosine for 15 min, and were then further grown for another 60 min after addition of 1.25 µM AbA. Sur7-mCherry was observed using a fluorescence microscope. The graphs show the fluorescence intensity profiles of Sur7-mCherry from the line scan (yellow line on midsection images of cells indicated by asterisks) along the PM. Scale bar: 10 µm. **c** *SUR7-mCherry* cells carrying an empty vector, YCpLG-*WSC1*, or YCpLG-*MID2* were cultured in raffinose medium until the log phase of growth, and galactose was added to the medium to a concentration of 2%. After incubation for 2 h, cells were treated with 1.25 µM AbA for 60 min. The fluorescence intensity profiles of Sur7-mCherry were determined as described in (**b**). Scale bar: 10 µm. **d** *SUR7-mCherry SLM1-GFP* cells were cultured and treated with AbA as described in (**b**). Sur7-mCherry and Slm1-GFP were observed using a fluorescence microscope. The graphs show the fluorescence intensity profiles of Sur7-mCherry (red line) and Slm1-GFP (green line) from a line scan along the PM. Scale bar: 2.5 µm. **e** *SUR7-mCherry SLM1-GFP* cells were cultured and treated with AbA as described in (**b**). Cortical localization of Sur7-mCherry and Slm1-GFP was observed using a fluorescence microscope. Quantitation of Sur7-mCherry and Slm1-GFP colocalization (Pearson's coefficient) was performed. Data represent mean ± s.d. Total number of cells analyzed in three independent experiments: control $n = 61$, AbA $n = 60$, edel $n = 55$, edel + AbA $n = 62$. Scale bar: 2.5 µm. **f** *SUR7-mCherry SLM1-GFP* cells carrying an empty vector or YCpLG-*PKC1^{R398P}* were cultured and treated with AbA as described in (**c**). Quantitation of Sur7-mCherry and Slm1-GFP colocalization was performed as described in (**e**). Total number of cells analyzed in three independent experiments: empty vector_control $n = 63$, vector_AbA $n = 61$, *PKC1^{R398P}*_control $n = 59$, *PKC1^{R398P}*_AbA $n = 64$. *$p < 0.05$; ***$p < 0.001$; ****$p < 0.0001$ (one-way ANOVA with Tukey's multiple comparisons test).

CFW on TORC2-Ypk1/2 signaling activation was examined. Pretreatment with CFW led to enhanced Mpk1 phosphorylation, indicating activation of the CWI pathway, and a decrease in the AbA-induced Ypk1/2 phosphorylation (Fig. 6a), suggesting that cell wall damage causes the inhibitory effect on TORC2-Ypk1/2 signaling. The inhibitory effect induced by CFW treatment was attenuated in *pil1Δ* cells (Fig. 6b), as was the case with edelfosine treatment or *MID2* overexpression. We next examined whether the inhibitory effect through the CWI pathway on TORC2-Ypk1/2 signaling is involved in an adaptive response to cell wall stress. If the inhibitory effect is important for the adaptive response to CFW, then *pil1Δ* cells would be expected to exhibit sensitivity to CFW. As shown in Fig. 6c, *PIL1* deletion increased CFW sensitivity, which was more pronounced at 35 °C than at 28 °C. Additionally, overexpression of *YPK1^{D242A}*, the constitutively active mutant allele of *YPK1*[19], also increased CFW sensitivity in wild-type cells (Fig. 6d). These results suggest that the downregulation of TORC2-Ypk1/2 signaling is involved in the proper adaptive response to CFW.

The yeast cell wall primarily comprises β-1,3-glucan, β-1,6-glucan, chitin, and mannoproteins, which are assembled into an extracellular matrix to maintain cell integrity, and is remodeled during cell growth by elongation and branching of newly synthesized polysaccharides. β-1,3-glucan is the most abundant cell wall component. *GAS1* encodes a glycosyl-phosphatidylinositol (GPI)-anchored protein with β-1,3-glucanosyltransferase activity that mediates the elongation and branching of β-1,3-glucan[54]. The *gas1Δ* cells have been shown to exhibit a decrease in β-1,3-glucan levels and weakened cell wall phenotypes, such as a hypersensitivity to cell wall-perturbing agents[55,56], and the CWI pathway is constitutively activated by *GAS1* deletion-induced cell wall defects[33]. Lastly, we examined whether the inhibitory effect on TORC2-Ypk1/2 signaling is exerted in *gas1Δ* cells with a constitutively activated state of the CWI pathway. *GAS1* deletion increased the Mpk1 phosphorylation levels, indicating the activation of the CWI pathway, and attenuated the AbA-induced Ypk1/2 phosphorylation (Fig. 6e). Additionally, the overexpression of *YPK1^{D242A}* led to growth defects in *gas1Δ* cells under unstressed conditions (Fig. 6f). These results suggest that the downregulation of TORC2-Ypk1/2 signaling is required for growth of *gas1Δ* cells, where the CWI pathway is constitutively activated.

## Discussion

TORC2-Ypk1/2 signaling senses PM dynamics and contributes to the maintenance of PM integrity[13,19,20,22,23]. Here, we identified a mechanism for the negative regulation of TORC2-Ypk1/2 signaling by the CWI pathway based on analysis using the lysophosphatidylcholine analog edelfosine. Edelfosine strongly inhibited TORC2-Ypk1/2 signaling without increasing mortality (Fig. 1), and edelfosine incorporated into PM caused CWI pathway activation without inducing cell wall damage (Fig. 2), which was involved in its inhibitory effect on TORC2-Ypk1/2 signaling (Fig. 3a, b). Edelfosine is known to alter PM microdomain organization and morphology[38–40]. It has been reported that sensors for CWI pathway, Wsc1 and Mid2, form specific microdomains at PM, and the extracellular cysteine-rich domain of Wsc1 contributes to its microdomain formation and signaling function[57]. Whether the formation of the sensor's microdomains triggers the CWI pathway is unclear, but edelfosine may be involved in the CWI pathway activation by altering those microdomains. TORC2 also resides in the MCT microdomain at PM, and its localization at PM is considered to be necessary to phosphorylate Ypk1/2 localized at PM via interaction with Slm1/2. Edelfosine did not disrupt the punctate pattern of MCT at PM (Supplementary Fig. 3b), and did not reduce the phosphorylation levels of Pkc1 at Ser1143 at Pkc1, another TORC2 target (Fig. 2b). Therefore, it was thought that the inhibitory effect of edelfosine on TORC2-Ypk1/2 signaling was unattributable to its influence on TORC2 itself.

The CWI pathway inhibited the activation of TORC2-Ypk1/2 signaling induced by membrane stress (Fig. 3) and simultaneously blocked eisosome disassembly and Slm1 release from eisosomes (Fig. 4). Since the inhibitory effect required eisosome structures (Fig. 5), we considered that the CWI pathway-induced inhibition of TORC2-Ypk1/2 signaling was exerted through the blockage of eisosome disassembly and Slm1 release. It has been reported that Slm1 has multiple phosphorylation sites, and that Slm1 phosphorylation is modulated by TORC2 or kinases regulated by phytosphingosine[25,58,59]. Slm1 is dephosphorylated following treatment with myriocin or AbA[21,59], and the dephosphorylation of Slm1 correlates with the phosphorylation of Ypk1, implying a relationship between the dephosphorylation of Slm1 and the activation of TORC2-Ypk1/2 signaling[21]. Edelfosine treatment enhanced the dephosphorylation of Slm1 as well as AbA treatment (Supplementary Fig. 4), suggesting that at least the dephosphorylation of Slm1 may not be involved in the release of Slm1 from eisosomes. Pkc1 in the CWI pathway was indispensable for the inhibitory effect on eisosome disassembly and Slm1 release, but the Mpk1 MAP kinase cascade was dispensable. Pkc1 has been reported to contribute to eisosome rearrangement under glucose starvation to promote the formation of stress granules, and eisosome rearrangement is controlled by Pil1 phosphorylations to which Pkc1 commits in an Mpk1-independent manner[60]. Ser230 of Pil1 has been shown to be the responsible phosphorylation site involved in the eisosome rearrangement regulated by Pkc1 under glucose starvation[60]. Previous phosphoproteomic analyses have indicated that Mpk1 also targets Ser230 and Thr233 of Pil1[52], and thus, Pil1 phosphorylation at Ser230 seems to be regulated by both Pkc1 and Mpk1. However, our results showed that the CWI pathway-induced inhibition of TORC2-Ypk1/2 signaling was unaffected by Pil1 mutation at Ser230 (Fig. 5d), suggesting that the effect of Pkc1 signaling on the phosphorylation levels of Pil1 at Ser230 was not involved in the blockage of eisosome disassembly. Pil1 is known to possess multiple phosphorylation sites[26], and therefore Pkc1 may block eisosome disassembly by regulating the phosphorylation levels of other sites in Pil1. Meanwhile, it has been reported that TORC2 phosphorylates some sites in the C-terminal region of Ypk1 in addition to the turn motif (Ser644) and hydrophobic motif (Thr662)[41], and the phosphorylation levels of these Ypk1 sites targeted by TORC2 other than Ser644 and Thr662 have been shown to be downregulated by Mpk1[61]. It has been suggested that this

**Fig. 5 | Involvement of eisosome structures in the inhibitory effect of the CWI pathway on TORC2-Ypk1/2 signaling. a** Wild-type (BY4741) and *pil1Δlsp1Δ* cells were cultured in SD medium until the log phase of growth and pretreated with 5 μM edelfosine for 15 min, and were then further grown for another 60 min after addition of 1.25 μM AbA. Phosphorylated Ypk1/2, Ypk1, phosphorylated Mpk1, or Mpk1 was detected as described in Fig. 3a. The intensity of immunoreactive bands was quantified using image analysis. The ratio of phosphorylated Ypk1/2 (p-Ypks) to total Ypk1 was normalized to that of the control sample of wild-type cells set as one. Data are from three or four independent experiments (mean ± s.d.). **b** Wild-type (BY4741) and *pil1Δlsp1Δ* cells carrying an empty vector or pRS413-*pGAL1-MID2* were cultured in raffinose medium until the log phase of growth, and galactose was added to the medium to a concentration of 2%. After incubation for 2 h, cells were treated with 1.25 μM AbA for 60 min. The phosphorylated Ypk1/2, Ypk1, phosphorylated Mpk1, or Mpk1 was detected as described in Fig. 3a. The intensity of the immunoreactive bands was quantified by image analysis. The ratio of phosphorylated Ypk1/2 (p-Ypks) to total Ypk1 was normalized to that of the control sample of wild-type cells carrying an empty vector set as one. Data are from four independent experiments (mean ± s.d.). **c** Wild-type (BY4741), *pil1Δ*, and *pil1Δlsp1Δ* cells were cultured in SD medium until the log phase of growth and treated with 5 μM edelfosine for 60 min, and then were spotted in 5-fold serial dilution onto SD agar plates with 0.03 μM AbA. The plates were scanned after incubation for 3 days at 28 °C. **d** *pil1Δ* cells carrying an empty vector pRS416-*PIL1-13myc* or pRS416-*PIL1*^S230A/T233A^-*13myc* were cultured in SD medium until the log phase of growth and pretreated with 5 μM edelfosine for 15 min, were then further grown for another 60 min after addition of 1.25 μM AbA. Phosphorylated Ypk1/2 or Ypk1 was detected as described in Fig. 1a. The intensity of the immunoreactive bands was quantified using image analysis. The ratio of phosphorylated Ypk1/2 (p-Ypks) to total Ypk1 was normalized to that of the control sample of *pil1Δ* cells carrying pRS416-*PIL1-13myc* cells set as one. Data are from three independent experiments (mean ± s.d.). *$p < 0.05$; **$p < 0.01$; ***$p < 0.001$; ****$p < 0.0001$ (one-way ANOVA with Tukey's multiple comparisons test).

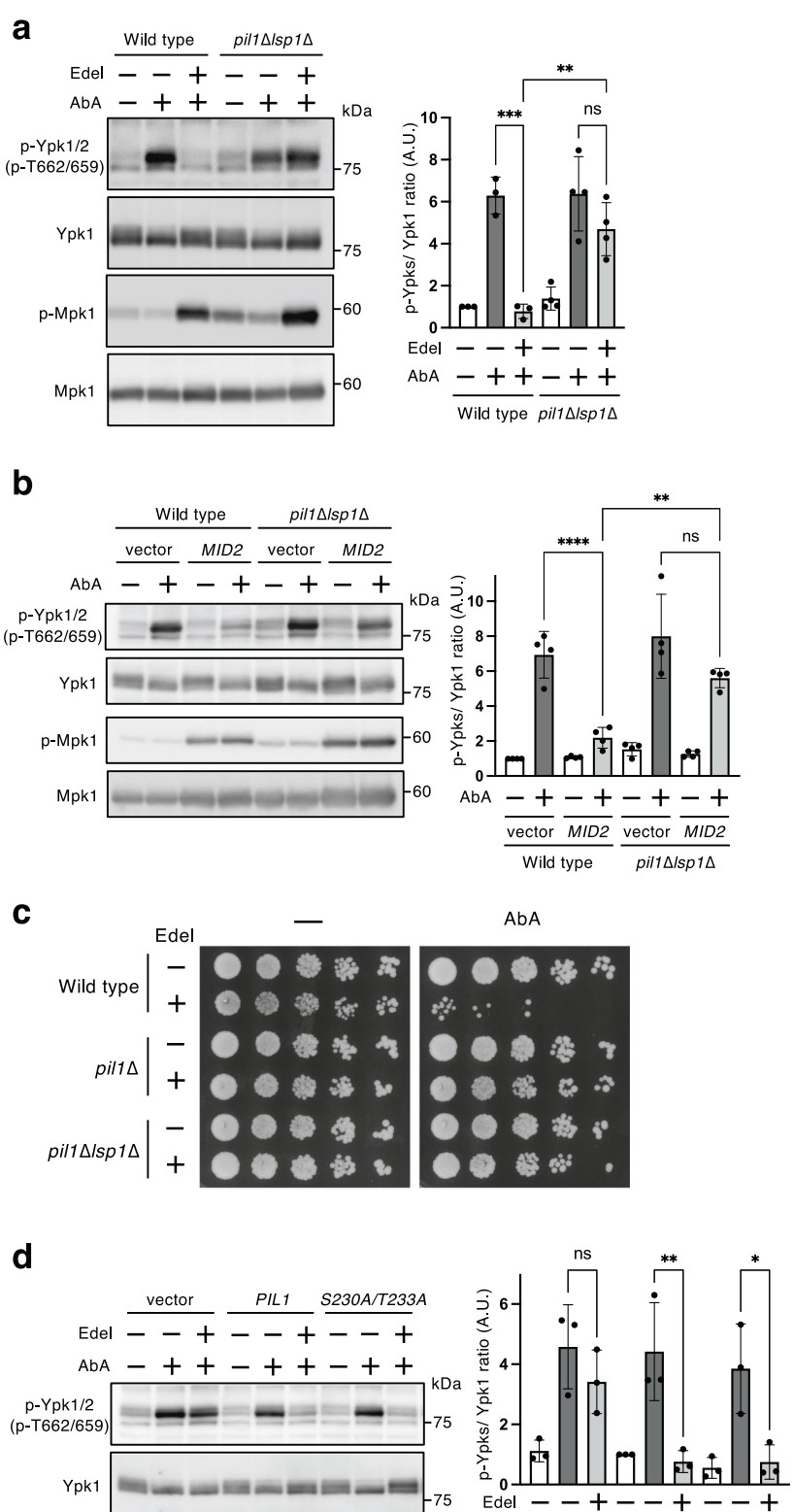

attenuation is due to Mpk1-mediated phosphorylation of Avo2, a TORC2 component, which promotes the dissociation of Avo2 from TORC2[61]. However, whether this Mpk1-dependent regulation affects the phosphorylation levels of Ypk1 at Thr662 has not been demonstrated[61]. Therefore, several TORC2-targeted phosphorylation sites of Ypk1 may be affected by Pkc1 signaling through different regulatory machinery that are either Mpk1-dependent or Mpk1-independent.

Our results show that the colocalization of Slm1 and eisosomes is enhanced by Pkc1 activation (Fig. 4e, f), suggesting that Pkc1 regulates the accumulation of Slm1 at eisosomes. The localization of Slm1 to PM requires its PH-domain that binds to PtdIns(4,5)$P_2$, and physical interactions between Slm1 and TORC2 components, Avo2 and Bit61, are thought to be involved in its localization to MCTs on PM[62]. On the other hand, the accumulation of Slm1 at eisosomes is thought to involve its PH-domain and

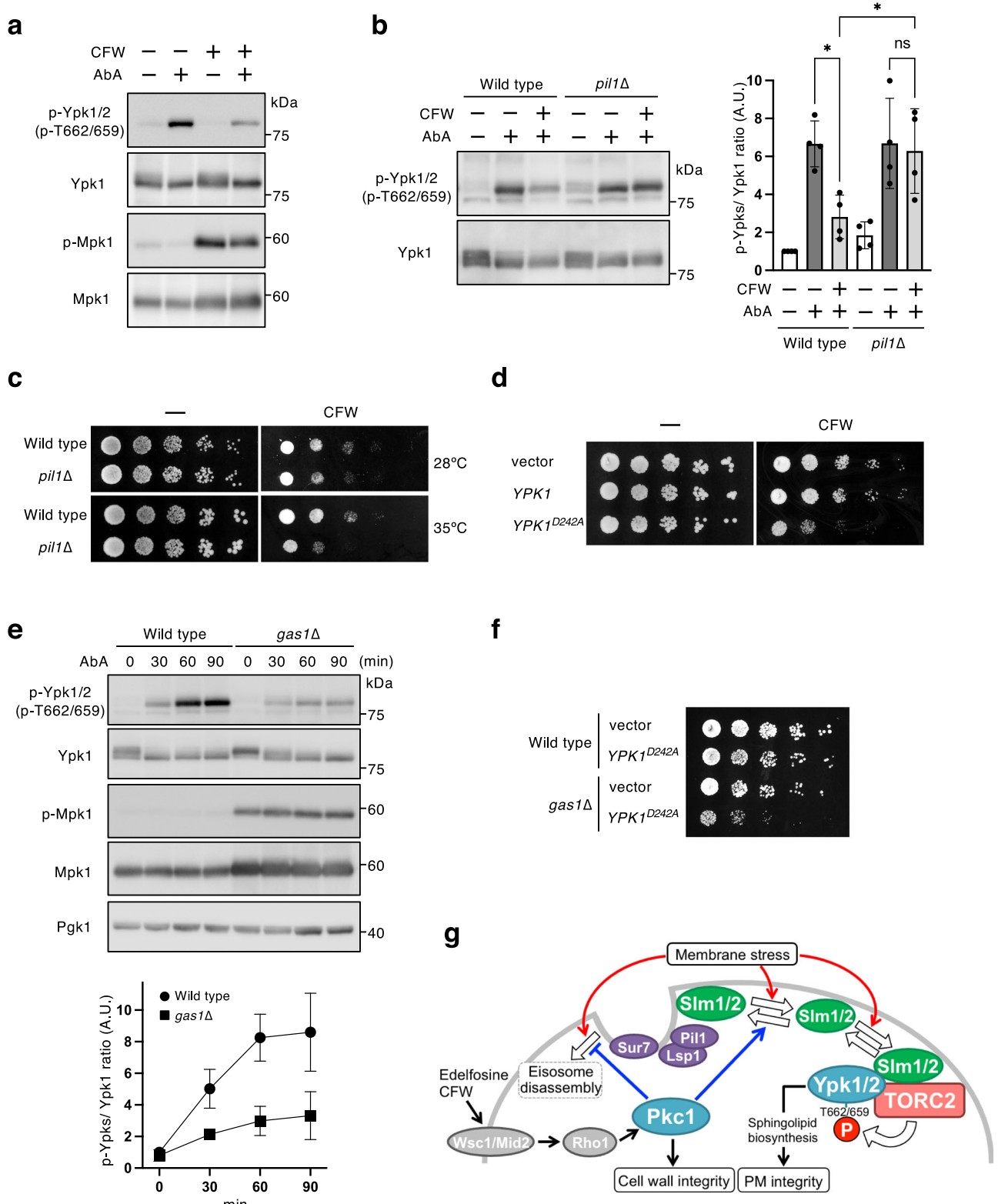

F-BAR domain that binds to membrane curvature[26], and PtdIns(4,5)$P_2$ has been shown to contribute to eisosome formation in addition to the accumulation of Slm1 at eisosomes[26,63,64]. Additionally, PtdIns(4,5)$P_2$ has been also reported to be involved in negative regulation of TORC2-Ypk1/2 signaling through a formation of PESs triggered by a decrease in plasma membrane tension[12]. We previously showed that activation of Pkc1 signaling by expression of $PKC1^{R398P}$ does not significantly change the

cellular levels of PtdIns(4,5)$P_2$[65]. Additionally, in microscopic analysis of the subcellular distribution of PtdIns(4,5)$P_2$ using GFP-2xPH$^{PLC\delta}$[66], an enrichment of GFP-2xPH$^{PLC\delta}$ at eisosomes and the PES formation were not observed in $PKC1^{R398P}$-expressing cells (our unpublished data). It is unlikely that an influence on the PtdIns(4,5)$P_2$ dynamics in PM is responsible for Slm1 accumulation at eisosomes by Pkc1, which is activated by the CWI pathway.

**Fig. 6 | The effect of cell wall stress on the inhibitory effect on TORC2-Ypk1/2 signaling. a, b** Wild-type (BY4741) and *pil1Δ* cells were cultured in SD medium until the log phase of growth and pretreated with 40 μg/ml CFW for 30 min, and were then further grown for another 60 min after the addition of 1.25 μM AbA. Phosphorylated Ypk1/2, Ypk1, phosphorylated Mpk1, or Mpk1 was detected as described in Fig. 3a. The intensity of immunoreactive bands was quantified using image analysis (**b**). The ratio of phosphorylated Ypk1/2 (p-Ypks) to total Ypk1 was normalized to that of the control sample of wild-type cells set as one. Data are from four independent experiments (mean ± s.d.). **c** Wild-type (BY4741) and *pil1Δ* cells were cultured in SD medium at 28 °C until the log phase of growth and serially diluted (1:5) using 0.85% NaCl solution. 5 μl each cell suspension was spotted onto SD agar plates with 400 μg/ml CFW and incubated at 28 °C and 35 °C for 3 days. **d** Wild-type (BY4741) cells carrying an empty vector, pRS423-*YPK1-3HA*, or pRS423-*YPK1^D242A^-3HA* were cultured and diluted as described in (**c**). 5 μl each cell suspension was spotted onto SD agar plates with 400 μg/ml CFW and incubated at 28 °C for 3 days. **e** Wild-type (BY4741) and *gas1Δ* cells were cultured in SD medium until the log phase of growth and pretreated with 5 μM edelfosine for 15 min, and treated with 1.25 μM AbA for the prescribed times. The phosphorylated Ypk1/2, Ypk1, phosphorylated Mpk1, or Mpk1 was detected as described in Fig. 3a. The intensity of the immunoreactive bands of phosphorylated Ypk1/2 and Ypk1was quantified by image analysis. The ratio of phosphorylated Ypk1/2 (p-Ypks) to that of total Ypk1 was normalized to the sample at 0 min set as one. Data are from three independent experiments (mean ± s.d.). **f** Wild-type (BY4741) and *gas1Δ* cells carrying an empty vector or pRS423-*YPK1^D242A^-3HA* were cultured and diluted as described in (**c**). 5 μl each cell suspension was spotted onto an SD agar plate and incubated at 28 °C for 3 days. **g** Schematic model for the negative regulation of TORC2-Ypk1/2 signaling by the CWI pathway. *$p < 0.05$ (one-way ANOVA with Tukey's multiple comparisons test).

PM integrity is essential for maintaining cellular homeostasis and cell growth. Lipids constituting eukaryotic PMs are primarily glycerophospholipids, sterols, and sphingolipids, and their levels change during cell growth and adaptation to environmental stresses, contributing to the maintenance of PM integrity[67,68]. The cell wall is constantly synthesized, remodeled, and degraded to maintain cell wall integrity under various extracellular conditions and environmental stresses[69]. During yeast cell growth, the synthesis and remodeling of the cell wall and the extension of the PM must be associated with an increase in cell volume, i.e., there should be a close relationship between cell wall integrity and PM integrity. In this study, we demonstrated that the CWI pathway contributing to cell wall integrity negatively regulated TORC2-Ypk1/2 signaling, which controls sphingolipid biosynthesis (Fig. 6g). Constitutive activation of Ypk1 signaling by expression of the active mutant *YPK1^D242A^* increased CFW sensitivity and caused a marked growth defect in *gas1Δ* cells (Fig. 6d, f). It has also been reported that expression of the active mutant *YPK2^D239A^* causes growth inhibition on agar plates containing sodium dodecyl sulfate (SDS), which induces plasma membrane and/or cell wall perturbation with activation of the CWI pathway[70]. These results may be due to the retention of high activity of Ypk1/2 signaling under conditions of cell wall damage, thereby upsetting the balance in relationship between cell wall integrity and PM integrity, which ensure the proper stress response and cell growth. Therefore, it is likely that the negative regulation of TORC2-Ypk1/2 signaling through the CWI pathway found in this study accounts for the regulation of this relationship.

Sphingolipid biosynthesis in yeast is carried out using ceramides, sphingolipid precursors, formed from very long-chain fatty acids (VLCFA) and sphingoid long-chain bases (LCBs), which are synthesized from two branched pathways[71]. The CWI pathway has been suggested to regulate VLCFA synthesis; a signaling pathway controlled by the guanine nucleotide exchange factor for Rho1, Rom2, negatively regulates the fatty acid elongation cycle via the phosphorylation of Elo2, fatty acid elongase[72]. Since the TORC2-Ypk1/2 signaling controls the synthesis of LCBs and ceramides through regulating serine palmitoyltransferase and ceramide synthase[18–21], our findings suggest that the CWI pathway negatively regulates sphingolipid biosynthesis by affecting multiple steps during ceramide synthesis through different mechanisms. Pkc1 has also been reported to target some proteins involved in phospholipid metabolism, including Pah1 (phosphatidate phosphatase)[73], Ura7 (CTP synthase)[74], and Opi1 (transcriptional repressor)[75]. The CWI pathway may control phospholipid metabolism in addition to sphingolipid metabolism through Pkc1 signaling, and this control of lipid metabolisms may contribute to the adaptive response to cell wall stress.

## Methods
### Media and reagents
Synthetic dextrose (SD) (2% glucose, 0.67% yeast nitrogen base without amino acids) or raffinose medium (2% raffinose, 0.67% yeast nitrogen base without amino acids) was used. Appropriate amino acids and bases were added as necessary. Aureobasidin A (630499) was obtained from TaKaRa Bio, Inc. (Otsu, Japan). Methylglyoxal (M0252), myriocin (M1177), amphotericin B (A4888), nystatin (N3503), filipin (F4767), and ethanolamine were purchased from Sigma-Aldrich (St. Louis, MO, USA). Edelfosine (3022) was purchased from Tocris Bioscience (Bristol, UK). Calcofluor White (158067) was purchased from MP Biomedicals (Santa Ana, CA, USA).

### Yeast strains
The yeast strains used are listed in Supplementary Table 1. The BY4741 background mutants, whose genes were deleted by the KanMX selection marker, come from the BY4741-based deletion mutant collection (Invitrogen, Carlsbad, CA, USA). The *wsc1Δmid2Δ* mutant was constructed by deleting *MID2* of the *wsc1Δ* mutant using PCR-based methods with a *his5^+^* selection marker[76]. The *pil1Δlsp1Δ* mutant was constructed by deleting *LSP1* of the *pil1Δ* mutant by a gene replacement method using the *Candida glabrata LEU2* gene (*CgLEU2*) as a selection marker[77]. TAP (tandem affinity purification)-tagged *ORM1* strain was used from the Yeast TAP-Fusion Library (Open Biosystems, Huntsville, AL, USA). To construct *PKC1* or *PKC1^L54S_4C/S^*, pRS305-*PKC1^Δ751-1151^* or pRS305-*PKC1^Δ751-1151^(L54S_4C/S)* was digested with StuI, and the linearized fragment was introduced into the locus of *PKC1*. The *PKC1^L54S_4C/S^* strain was isolated and cultured under medium conditions containing 1 M sorbitol. To add an mCherry tag at the C terminus of Sur7 or Lsp1, pRS306-*SUR7-mCherry* or pRS30Nat-*LSP1-mCherry* was digested with MfeI or ClaI, and the linearized fragment was introduced into the locus of *SUR7* or *LSP1*, respectively. To add a GFP tag at the C terminus of Slm1, pRS30Gen-*SLM1-GFP* was digested with SphI, and the linearized fragment was introduced into the locus of *SLM1*. The *AVO3-3GFP* allele was introduced into the *AVO3* locus using the integration plasmid pRS305-*AVO3-3GFP*[36].

### Plasmids
The plasmids used in this study are summarized in Supplementary Table 2. The overexpressed plasmids with the *GAL1* promoter, YCpLG-*WSC1*, and YCpLG-*MID2*, were constructed as follows: the ORF of *WSC1* or *MID2* and each 3′ untranslated region were amplified using the following primers: 5′-ATTGGATCCATGAGACCGAACAAAACAAGTCTGCT-3′ and 5′-TATGTCGACGCATCCTATCACATAATCAATCCTTG-3′, or 5′-AATGGATCCATGTTGTCTTTCACAACCAAGAATAG-3′ and 5′-TATGTCGACGAGAGAATATCTCCTTATGCTTCTAC-3′. Each PCR product was digested with BamHI and SalI, and the resultant fragments were introduced into the BamHI and SalI sites of YCpLG, respectively. The fragment containing *pGAL1-MID2* obtained by digestion of YCpLG-*MID2* with PvuII was cloned into the PvuII sites of pRS413 to generate pRS413-*pGAL1-MID2*. The plasmid pRS416-*HA-RHO1* or pRS416-*HA-RHO1^Q68L^* was constructed as follows: the ORF of *RHO1* or *RHO1^Q68L^* and each 3′ untranslated region were amplified by PCR with plasmid pYO701 or pYO962[78] as a template using the following primers: 5′-ATACTGCAGGGATGTCACAACAAGTTGGTAACAG-3′ and 5′-AAAGAGCTCGCATACGTACATACAATGAGAAAAGG-3′. Each PCR product was digested with PstI and SacI, and the resultant fragment was introduced into the PstI and SacI sites of pRS414-*pGAL-HA* (laboratory stock), a plasmid for HA-

tagged protein expression under the *GAL1* promoter. The fragment containing *pGAL1-HA-RHO1* or *pGAL1-HA-RHO1^{Q68L}* obtained by digestion of pRS414-*pGAL-HA-RHO1* or pRS414-*pGAL-HA-RHO1^{Q68L}* with XhoI and SacI was cloned into the XhoI and SacI sites of pRS416 to generate pRS416-*pGAL-HA-RHO1* or pRS416-*pGAL-HA-RHO1^{Q68L}*. The plasmid pRS416-*PIL1-13myc* was constructed as follows: the 5′ untranslated region of *PIL1* and the ORF of *PIL1* were amplified using the following primers: 5′-AAAGCGGCCGCTAACCACTGATTCAGAGTTCCAGA-3′ and 5′-AAAGGATCCGAGCTGTTGTTTGTTGGGGAAGAGAC-3′. The PCR product was digested with NotI and BamHI, and the resultant fragment was introduced into the NotI and BamHI sites of pRS416-*13myc* (laboratory stock). The plasmid pRS416-*PIL1^{S230A/T233A}-13myc* was constructed using a KOD -Plus- Mutagenesis Kit (TOYOBO) with pRS416-*PIL1-13myc* as a template. The plasmid pRS423-*YPK1-3HA* was constructed as follows: the 5′ untranslated region of *YPK1* and the ORF of *YPK1* were amplified using the following primers: 5′-CTGTCGACCTCCATCTCTGAAAATCAA GGC-3′ and 5′-CACAGATCTCCTCTAATGCTTCTACCTTGC-3′. The PCR product was digested with SalI and BglII, and the resultant fragment was introduced into the SalI and BglII sites of pRS424-*PKC1-3HA*[16]. The fragment containing *YPK1-3HA* obtained by digesting pRS424-*YPK1-3HA* with XhoI and SacI was cloned into the XhoI and SacI sites of pRS423 to generate pRS423-*YPK1-3HA*. The plasmid pRS423-*YPK1^{D242A}-3HA* was constructed using the KOD -Plus- Mutagenesis Kit with pRS423-*YPK1-3HA* as a template. To construct the integration plasmid pRS305-*PKC1^{Δ751-1151}* or pRS305-*PKC1^{Δ751-1151}(L54S_4 C/S)*, the C-terminal region of *PKC1* and ORF of *3GFP* in pRS305-*PKC1-3GFP* or pRS305-*PKC1^{4C/S}-3GFP*[49] were deleted using the KOD-Plus-Mutagenesis Kit. The resultant plasmid pRS305-*PKC1^{Δ751-1151}(4C/S)* was additionally point mutated to obtain the pRS305-*PKC1^{Δ751-1151}(L54S_4C/S)*. The plasmid pRS30Gen-*SLM1-GFP* was constructed as follows: the C-terminal region of *SLM1* was amplified using the following primers: 5′-AATTCTAGACCAGT ATCATGAATCCCTAGCATCCA-3′ and 5′-TATCTCGAGCTGATTT GAAATCGTTTATTGGAGTT-3′. The PCR product was digested with XbaI and XhoI, and the resultant fragment was introduced into the XbaI and XhoI sites of pRS30Gen-*GFP* (laboratory stock). The plasmid pRS306-*SUR7-mCherry* and pRS30Nat-*LSP1-mCherry* were constructed as follows: the C-terminal region of *SUR7* or *LSP1* was amplified using the following primers: 5′-AAAGAGCTCTCGGATACCTGCACAAGCAATTTGGC-3′ and 5′-ATACTCGAGCAACAGAGACATCGTCCGGGCGCTCG-3′ or 5′-TTAGAGCTCTGGAAGTTGTTGCCAGCGAACGCCGT-3′ and 5′-AAACTCGAGCGATGTTTTCAGAACCGGAGGTATGA-3′. Both PCR products were digested with SacI and XhoI, and the resultant fragments were introduced into the SacI and XhoI sites of pRS306-*mCherry* or pRS30Nat-*mCherry* (laboratory stock), respectively.

### Western blotting
The cells were cultured until an $A_{610}$ of 0.3–0.5 was obtained. Cells harvested by centrifugation were resuspended in 200 μl of 20% trichloroacetic acid with glass beads and disrupted using Micro Smash MS-100 (TOMY, Tokyo, Japan). Supernatants were obtained, and glass beads were rinsed with 400 μl of 5% trichloroacetic acid (sample's final concentration of trichloroacetic acid was 10%), and then the samples were incubated for another 15 min on ice. Samples were centrifuged at 4 °C for 10 min, and pellets were rinsed using ice-cold acetone. Following centrifugation, pellets were dried and resuspended using 2× Laemmli sample buffer. The total protein concentration was determined using the RC DC™ Protein Assay (#5000122; Bio-Rad Laboratories, Hercules, CA, USA). The samples were subjected to SDS-PAGE, and the separated proteins were transferred onto a polyvinylidene difluoride membrane (Millipore, Burlington, MA, USA). Dr. Tatsuya Maeda of Hamamatsu University School of Medicine kindly donated the anti-phosphorylated Ypk1^{T662}/Ypk2^{T659} (1:25,000)[21], anti-Ypk1 (1:3000, sc12054; Santa Cruz Biotechnology, Dallas, TX, USA), anti-phospho-p44/42 MAP kinase (1:5000, #9101; Cell Signaling Technology, Danvers, MA, USA), anti-Mpk1 (1:3,000, sc6803; Santa Cruz Biotechnology), anti-Pgk1 (1:10,000, #A6457; Molecular Probes, Eugene, OR, USA),

anti-phosphorylated Pkc1^{S1143} (1:3000)[16], and anti-Pkc1 (1:2000)[16], were used as primary antibodies. The Orm1-TAP was detected using peroxidase anti-peroxidase soluble complex (PAP) antibody (1:8,000, P1291; Sigma-Aldrich). Immunoreactive bands were visualized using Immobilon Western chemiluminescent horseradish peroxidase substrate (Millipore) using a Fuji Las 4000 Mini System (Fujifilm, Tokyo, Japan). Size markers were estimated using digitized image of pre-stained protein markers transferred to the polyvinylidene difluoride membrane. The intensity of the immunoreactive bands was quantified using ImageJ.

### Spot assay
Cells were cultured in SD medium until the early log phase of growth and then diluted to an $A_{610}$ value of 0.1 using sterilized 0.85% NaCl solution. The cells were serially diluted (1:5) with sterilized 0.85% NaCl solution, and aliquots (5 μL) were spotted onto SD agar plates.

### Microscopy
Microscopic analysis using a micro slide glass was conducted using a BX63 fluorescence microscope (OLYMPUS, Tokyo, Japan) equipped with an ORCA-Fusion digital camera (Hamamatsu Photonics, Shizuoka, Japan). To analyze the fluorescence intensity profiles of Sur7-mCherry and Slm1-GFP, each fluorescence intensity along the PM on midsection images was calculated using cellSens software (OLYMPUS). The colocalization of Sur7-mCherry and Slm1-GFP was analyzed using the cellSens software with microscopic images of the cell surface and is represented by the Pearson's correlation coefficient.

### Statistics and reproducibility
Statistical analysis was performed using GraphPad Prism 9. Sample sizes are indicated in the figure legends. Data are presented as mean and standard deviation. The statistical significance of differences was assessed using one-way ANOVA with Tukey's multiple comparison test. Differences were considered significant at $p < 0.05$.

### Reporting summary
Further information on research design is available in the Nature Portfolio Reporting Summary linked to this article.

### Data availability
The source data of the graphs are provided as a Supplementary Data file. Uncropped western blot images are shown in Supplementary Fig. 5. The data that support the findings of this study are available from the corresponding authors upon reasonable request.

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

## Acknowledgements

We are grateful to Drs. T. Maeda, and T. Takahara for providing the anti-phosphorylated $Ypk1^{T662}/Ypk2^{T659}$ antibody; Drs. D.E. Levin, Y. Ohya, J. Thorner, and the National Bio-Resource Project (NBRP) of MEXT, Japan, for providing plasmids and yeast strains. We thank K. Ikeda and Dr. S-P. Ng for providing the technical support and valuable comments on the manuscript. This work was partly supported by Japan Society for the Promotion of Science (JSPS) KAKENHI Grants (number 22K05559 to W.N., number 21H02103 to Y.I.), and a Lotte Foundation, Japan, Shigemitsu Prize (to W.N.).

## Author contributions

Conceptualization: W.N. and Y.I.; Data curation: W.N.; Investigation: W.N.; Writing – original draft: W.N. and Y.I.; Writing – review & editing: W.N. and Y.I.; Resources: W.N. and Y.I.; Project administration: W.N.; Supervision: Y.I.; Visualization: W.N.; Funding acquisition: W.N. and Y.I.

## Competing interests

The authors declare no competing interests.
