## [Peer review file · Communications Biology]

REVIEWERS' COMMENTS:

Reviewer #1 (Remarks to the Author):

The authors addressed all of my comments, and I suggest to accept the revised version.

Reviewer #3 (Remarks to the Author):

The authors addressed all major concerns and therefore I support the publication of this manuscript.

To reviewer #1

Thank you very much for your critical review and fruitful suggestion on our manuscript. Your comments are highly appreciated. We feel the comments have helped us significantly to improve the paper. Please find below a point by point response to your comments. Modified sentences, as well as the added part, are shown in red font in the revised manuscript.

In this manuscript, Nomura and Inoue provide interesting findings on crosstalk between the plasma membrane integrity (PMI) and cell wall integrity (CWI) pathways in yeast.

They found that edelfosine, which activates the CWI pathway without cell wall damage, inhibits activation of the TORC2-Ypk1/2 signaling pathway associated with inhibition of sphingolipid synthesis by Aureobasidin A. Furthermore, they found that the TORC2-Ypk1/2 signaling pathway was similarly inhibited under conditions that activate other CWI pathways such as CFW treatment or in gas1Δ cells. They also found that this inhibition of the TORC2-Ypk1/2 signaling pathway by activation of the CWI pathway is not due to direct inhibition of TORC2 or Ypk1/2, but rather to inhibition of disassembly of the eisosome which is upstream of the TORC2-Ypk1/2 signaling pathway. Although there have been several reports on the relationship between eisosomes, the CWI pathway, and the TORC2-Ypk1/2 signaling pathway, this manuscript provides new insights that will be of interest to general readers in this field. Overall, the manuscript is interesting, and the data support the main conclusions and are worthy of consideration for publication but would be even better if the following comments were improved.

Major comment

In Fig. 4, it is interesting that edelfosine inhibits eisosome disassembly and release of Slm1 from eisosomes via activation of the CWI pathway. This phenomenon is dependent on Pkc1, not on Mpk1 downstream of the CWI MAP kinase cascade. It was reported that Slm1 is dephosphorylated upon inhibition of sphingolipid synthesis (Ishino et al., FEBS J, 2022). How does edelfosine affect the phosphorylation of Slm1? Also, does edelfosine affect the level of Sur7 phosphorylation in the same way? Proving these questions may help us understand how Pkc1 regulates eisosome disassembly.

We thank your kind suggestion regarding the levels of Slm1 phosphorylation. It has been reported that Slm1 has multiple phosphorylation sites, and that Slm1 phosphorylation is modulated by TORC2 or kinases regulated by phytosphingosine (EMBO J. 23:3747-3757, 2004; Mol. Cell. Biol. 26:4729-4745, 2006; Mol. Cell. Biol. 27:633-650, 2007). The levels of Slm1 phosphorylation are transiently diminished following heat shock stress, suggesting a link between Slm1 activity and organizations of the actin cytoskeleton (EMBO J. 23:3747-3757, 2004). As the reviewer has pointed out, Slm1 is also dephosphorylated following treatment with inhibitors of sphingolipid synthesis, myriocin and AbA (Mol. Cell. Biol. 27:633-650, 2007; FEBS J. 289:457-472, 2022). Ishino et al. (FEBS J. 2022), have been shown that the dephosphorylation of

Slm1 induced by the inhibition of sphingolipid synthesis correlates with the phosphorylation of Ypk1, implying a relationship between the dephosphorylation of Slm1 and the activation of TORC2-Ypk1/2 signaling.

According to the reviewer's suggestion, we examined the effect of edelfosine on the levels of Slm1 phosphorylation. As shown in Suppl. Fig. 4 in the revised manuscript, the dephosphorylation of Slm1 induced following treatment with AbA was also observed in the experimental conditions using our yeast strain, and in addition, edelfosine caused the dephosphorylation of Slm1 as well. It has been reported that the calcineurin phosphatase interacts with Slm1/2 and dephosphorylates them (Mol. Cell. Biol. 26:4729-4745, 2006). We examined the involvement of calcineurin in the inhibitory effect of edelfosine on TORC2-Ypk1/2 signaling as shown in an attached figure below. The inhibitory effect of edelfosine on Ypk1/2 phosphorylation was not alleviated by deletion of Cnb1, a regulatory subunit of calcineurin. Since AbA and edelfosine had the opposing effects on the Slm1 localization to eisosomes, we believe that the edelfosine-induced blockage of Slm1 release from eisosomes is not due to a decrease in the levels of Slm1 phosphorylation. We added and modified relevant sentences in the revised manuscript (lines 385-393: **“It has been reported that Slm1 has multiple phosphorylation sites, and that Slm1 phosphorylation is modulated by TORC2 or kinases regulated by phytosphingosine (Audhya *et al.*, 2004; Bultynck *et al.*, 2006; Daquinag *et al.*, 2007). Slm1 is dephosphorylated following treatment with myriocin or AbA (Daquinag *et al.*, 2007; Ishino *et al.*, 2022), and the dephosphorylation of Slm1 correlates with the phosphorylation of Ypk1, implying a relationship between the dephosphorylation of Slm1 and the activation of TORC2-Ypk1/2 signaling (Ishino *et al.*, 2022). Edelfosine treatment enhanced the dephosphorylation of Slm1 as well as AbA treatment (Suppl. Fig. 4), suggesting that at least the dephosphorylation of Slm1 may not be involved in the release of Slm1 from eisosomes.”**).

We examined the levels of Sur7 phosphorylation as shown in an attached figure below. The phosphorylated states of Sur7 were not observed in our experimental conditions. The precise mechanism underlying how Pkc1 contributes to the inhibition of eisosome disassembly and release of Slm1 from eisosomes, including the possibility that Pkc1 targets eisosome-related proteins as suggested by the reviewer, is very interesting and would be our next task.

Minor comments

1) *It should be indicated in the figure that the +/- in Fig. 5 C indicates with or without edelfosine treatment.*

We thank the reviewer for this indication. Following the reviewer’s comment, we modified the Fig. 5C.

2) *p11, line 337-338: The reason for treating CFW at 35°C in Fig. 6C should be mentioned.*

Following the reviewer’s comment, we modified relevant sentence in the revised manuscript (lines 337-338: “As shown in Fig. 6C, *PIL1* deletion increased CFW sensitivity, which was more pronounced at 35°C than at 28°C.”)

3) *In Fig. 6, constant activation of Ypk1 causes growth inhibition under conditions that activate the CWI pathway, as was already shown by Sakata et al. in Mol Microbiol, 2022, where constant activation of Ypk2 also causes growth inhibition in SDS that activates the CWI pathway. This should be addressed in the Discussion.*

Following the reviewer’s comment, we added sentence in the revised manuscript (lines 449-452: “It has also been reported that expression of the active mutant *YPK2^{D239A}* causes growth inhibition on agar plates containing sodium dodecyl sulfate (SDS), which induces plasma membrane and/or cell wall perturbation with activation of the CWI pathway (Sakata et al., 2022).”)

To Reviewer 2

We sincerely appreciate your positive evaluation of our manuscript and your constructive comments that have markedly improved the quality of the paper. Please find below a point by point response

to your comments. Modified sentences, as well as the added part, are shown in red font in the revised manuscript.

The authors analyzed the effect of the lysophospholipid-like molecule edelfosine on the TORC2-Ypk1/2 signaling pathway. The findings show that edelfosine activates the kinase Pkc1 which is part of the CWI stress response pathway. Active Pkc1 inhibits TORC2-Ypk1/2 most likely by affecting eisosome structure and thus stabilizing the association of Slm1/2 with these eisosomes. This downregulation of TORC2 signaling seems to be important for the proper response of yeast to cell wall stress.

This study has been well conducted and the data support the general conclusions. However, the study relies heavily on the use of drugs and combination of drugs and therefore the relevance of the findings for normal physiological conditions of yeast is not always clear. Furthermore, previous studies in this field have not been sufficiently incorporated into the design and interpretation of the experiments. Therefore, I suggest the following additions/changes:

1) A previous study showed that the addition of the palmitoylcarnitine (PalmC), a surfactant-like molecule similar to that of edelfosine, causes loss of plasma membrane tension (by intercalating into the membrane), clustering of PI4,5P2, localization of TORC2 to these clusters and the resulting shutdown of TORC2-Ypk1/2 signaling (Nat Cell Biol 2018, 20(9)). In this context, the shutdown of sphingolipid synthesis makes sense since the plasma membrane experiences low tension (too many lipids). The question is: does edelfosine act similarly? Does edelfosine cause loss of plasma membrane tension? Does PalmC cause activation of the CWI pathway? Loss of plasma membrane tension could explain the increased stabilization of the eisosomes and the increased localization of Slm1/2 to eisosomes.

We thank the reviewer for this constructive suggestion. As the reviewer has pointed out, PalmC reduces a plasma membrane tension and emerges an invaginated plasma membrane structure in which PtdIns(4,5)P₂ accumulates (called PES) (Nat Cell Biol 2018, 20(9)). The formation of PESs is also observed by hyper-osmotic shock that result in a decrease in plasma membrane tension (Nat Cell Biol 2018, 20(9)). Indeed, structurally edelfosine and PalmC are similar; however, no clear formation of PESs was observed following treatment with edelfosine in our experimental conditions as shown in an attached figure below. Additionally, an overexpression of active Pkc1 also did not induce the PES formation. Meanwhile, it is generally understood that the CWI pathway is activated under conditions that increase the plasma membrane tension, such as hypo-osmotic shock, and conversely, hyper-osmotic shock, which reduces the plasma membrane tension and induces PES formation as well as PalmC, is known to depress the activity of CWI pathway (J. Biol. Chem. 270:30157-30161, 1995; J. Biol. Chem. 277:21278-21284, 2002). Therefore, we believe that edelfosine does not cause loss of the plasma membrane tension and inhibits TORC2-Ypk1/2 signaling through a different mechanism from that of PalmC, i.e., through activation of CWI pathway. The mechanism underlying how edelfosine activates the CWI pathway is very interesting, and we are focusing

on the effect of edelfosine on Wsc1 and Mid2 (sensors for CWI pathway), which would be the subject of future research.

We added and modified relevant sentences in the revised manuscript (lines 427-429: “**Additionally, PtdIns(4,5)P₂ has been also reported to be involved in negative regulation of TORC2-Ypk1/2 signaling through a formation of PESs triggered by a decrease in plasma membrane tension (Riggi *et al.*, 2018).**”, lines 431-434: “Additionally, in microscopic analysis of the subcellular distribution of PtdIns(4,5)P₂ using GFP-2xPH^{PLCd} (Stefan *et al.*, 2002), an enrichment of GFP-2xPH^{PLCd} at eisosomes **and the PES formation were** not observed in *PKC1^{R398P}*-expressing cells (data not shown).”).

2) To test for physiological relevance, the authors should include in their studies hyperosmotic stress, which is known to induce the CWI pathway. Again, this might be caused by the loss of plasma membrane tension (cells shrink and PM tension is lost). Does hyperosmotic shock cause the shutdown of TORC2-Ypk1/2 signaling via *Pkc1*?

As mentioned above, the CWI pathway in *S. cerevisiae* is basically considered to be activated when the plasma membrane tension is increased. On the other hand, as the reviewer has pointed out, there is indeed a study showing that the CWI pathway is weakly and transiently activated in response to hyper-osmotic shock through the high-osmolarity glycerol (HOG) pathway that produces the compatible osmolyte glycerol (FEBS Lett. 579:6186-6190, 2005). The activation of the CWI pathway induced by hyper-osmotic shock occurs after a considerable delay, as the activity of HOG pathway returns to basal levels. It has been suggested that the hyper-osmotic shock-induced activation of the CWI pathway may be the result of an overshoot in intracellular glycerol concentrations, which would increase turgor pressure (Fungal Biol. 124:361-367, 2020). Therefore, the loss of plasma membrane tension is not likely to be a direct cause of the CWI pathway activation, and we believe that the inhibitory effect of *Pkc1* on the TORC2-Ypk1/2 signaling is not exerted by loss of the plasma membrane tension.

REVIEWERS' COMMENTS:

Reviewer #1 (Remarks to the Author):

The authors addressed all of my comments, and I suggest to accept the revised version.

Reviewer #3 (Remarks to the Author):

The authors addressed all major concerns and therefore I support the publication of this manuscript.

To reviewer #1

Thank you very much for your critical review and fruitful suggestion on our manuscript. Your comments are highly appreciated. We feel the comments have helped us significantly to improve the paper. Please find below a point by point response to your comments. Modified sentences, as well as the added part, are shown in red font in the revised manuscript.

In this manuscript, Nomura and Inoue provide interesting findings on crosstalk between the plasma membrane integrity (PMI) and cell wall integrity (CWI) pathways in yeast.

They found that edelfosine, which activates the CWI pathway without cell wall damage, inhibits activation of the TORC2-Ypk1/2 signaling pathway associated with inhibition of sphingolipid synthesis by Aureobasidin A. Furthermore, they found that the TORC2-Ypk1/2 signaling pathway was similarly inhibited under conditions that activate other CWI pathways such as CFW treatment or in gas1Δ cells. They also found that this inhibition of the TORC2-Ypk1/2 signaling pathway by activation of the CWI pathway is not due to direct inhibition of TORC2 or Ypk1/2, but rather to inhibition of disassembly of the eisosome which is upstream of the TORC2-Ypk1/2 signaling pathway. Although there have been several reports on the relationship between eisosomes, the CWI pathway, and the TORC2-Ypk1/2 signaling pathway, this manuscript provides new insights that will be of interest to general readers in this field. Overall, the manuscript is interesting, and the data support the main conclusions and are worthy of consideration for publication but would be even better if the following comments were improved.

Major comment

In Fig. 4, it is interesting that edelfosine inhibits eisosome disassembly and release of Slm1 from eisosomes via activation of the CWI pathway. This phenomenon is dependent on Pkc1, not on Mpk1 downstream of the CWI MAP kinase cascade. It was reported that Slm1 is dephosphorylated upon inhibition of sphingolipid synthesis (Ishino et al., FEBS J, 2022). How does edelfosine affect the phosphorylation of Slm1? Also, does edelfosine affect the level of Sur7 phosphorylation in the same way? Proving these questions may help us understand how Pkc1 regulates eisosome disassembly.

We thank your kind suggestion regarding the levels of Slm1 phosphorylation. It has been reported that Slm1 has multiple phosphorylation sites, and that Slm1 phosphorylation is modulated by TORC2 or kinases regulated by phytosphingosine (EMBO J. 23:3747-3757, 2004; Mol. Cell. Biol. 26:4729-4745, 2006; Mol. Cell. Biol. 27:633-650, 2007). The levels of Slm1 phosphorylation are transiently diminished following heat shock stress, suggesting a link between Slm1 activity and organizations of the actin cytoskeleton (EMBO J. 23:3747-3757, 2004). As the reviewer has pointed out, Slm1 is also dephosphorylated following treatment with inhibitors of sphingolipid synthesis, myriocin and AbA (Mol. Cell. Biol. 27:633-650, 2007; FEBS J. 289:457-472, 2022). Ishino et al. (FEBS J. 2022), have been shown that the dephosphorylation of

Slm1 induced by the inhibition of sphingolipid synthesis correlates with the phosphorylation of Ypk1, implying a relationship between the dephosphorylation of Slm1 and the activation of TORC2-Ypk1/2 signaling.

According to the reviewer's suggestion, we examined the effect of edelfosine on the levels of Slm1 phosphorylation. As shown in Suppl. Fig. 4 in the revised manuscript, the dephosphorylation of Slm1 induced following treatment with AbA was also observed in the experimental conditions using our yeast strain, and in addition, edelfosine caused the dephosphorylation of Slm1 as well. It has been reported that the calcineurin phosphatase interacts with Slm1/2 and dephosphorylates them (Mol. Cell. Biol. 26:4729-4745, 2006). We examined the involvement of calcineurin in the inhibitory effect of edelfosine on TORC2-Ypk1/2 signaling as shown in an attached figure below. The inhibitory effect of edelfosine on Ypk1/2 phosphorylation was not alleviated by deletion of Cnb1, a regulatory subunit of calcineurin. Since AbA and edelfosine had the opposing effects on the Slm1 localization to eisosomes, we believe that the edelfosine-induced blockage of Slm1 release from eisosomes is not due to a decrease in the levels of Slm1 phosphorylation. We added and modified relevant sentences in the revised manuscript (lines 385-393: **“It has been reported that Slm1 has multiple phosphorylation sites, and that Slm1 phosphorylation is modulated by TORC2 or kinases regulated by phytosphingosine (Audhya *et al.*, 2004; Bultynck *et al.*, 2006; Daquinag *et al.*, 2007). Slm1 is dephosphorylated following treatment with myriocin or AbA (Daquinag *et al.*, 2007; Ishino *et al.*, 2022), and the dephosphorylation of Slm1 correlates with the phosphorylation of Ypk1, implying a relationship between the dephosphorylation of Slm1 and the activation of TORC2-Ypk1/2 signaling (Ishino *et al.*, 2022). Edelfosine treatment enhanced the dephosphorylation of Slm1 as well as AbA treatment (Suppl. Fig. 4), suggesting that at least the dephosphorylation of Slm1 may not be involved in the release of Slm1 from eisosomes.”**).

We examined the levels of Sur7 phosphorylation as shown in an attached figure below. The phosphorylated states of Sur7 were not observed in our experimental conditions. The precise mechanism underlying how Pkc1 contributes to the inhibition of eisosome disassembly and release of Slm1 from eisosomes, including the possibility that Pkc1 targets eisosome-related proteins as suggested by the reviewer, is very interesting and would be our next task.

Minor comments

1) It should be indicated in the figure that the +/- in Fig. 5 C indicates with or without edelfosine treatment.

We thank the reviewer for this indication. Following the reviewer's comment, we modified the Fig. 5C.

2) p11, line 337-338: The reason for treating CFW at 35°C in Fig. 6C should be mentioned.

Following the reviewer's comment, we modified relevant sentence in the revised manuscript (lines 337-338: "As shown in Fig. 6C, *PIL1* deletion increased CFW sensitivity, which was more pronounced at 35°C than at 28°C.")

3) In Fig. 6, constant activation of *Ypk1* causes growth inhibition under conditions that activate the CWI pathway, as was already shown by Sakata *et al.* in *Mol Microbiol*, 2022, where constant activation of *Ypk2* also causes growth inhibition in SDS that activates the CWI pathway. This should be addressed in the Discussion.

Following the reviewer's comment, we added sentence in the revised manuscript (lines 449-452: "It has also been reported that expression of the active mutant *YPK2^{D239A}* causes growth inhibition on agar plates containing sodium dodecyl sulfate (SDS), which induces plasma membrane and/or cell wall perturbation with activation of the CWI pathway (Sakata *et al.*, 2022).")

To Reviewer 2

We sincerely appreciate your positive evaluation of our manuscript and your constructive comments that have markedly improved the quality of the paper. Please find below a point by point response

to your comments. Modified sentences, as well as the added part, are shown in red font in the revised manuscript.

The authors analyzed the effect of the lysophospholipid-like molecule edelfosine on the TORC2-Ypk1/2 signaling pathway. The findings show that edelfosine activates the kinase Pkc1 which is part of the CWI stress response pathway. Active Pkc1 inhibits TORC2-Ypk1/2 most likely by affecting eisosome structure and thus stabilizing the association of Slm1/2 with these eisosomes. This downregulation of TORC2 signaling seems to be important for the proper response of yeast to cell wall stress.

This study has been well conducted and the data support the general conclusions. However, the study relies heavily on the use of drugs and combination of drugs and therefore the relevance of the findings for normal physiological conditions of yeast is not always clear. Furthermore, previous studies in this field have not been sufficiently incorporated into the design and interpretation of the experiments. Therefore, I suggest the following additions/changes:

1) A previous study showed that the addition of the palmitoylcarnitine (PalmC), a surfactant-like molecule similar to that of edelfosine, causes loss of plasma membrane tension (by intercalating into the membrane), clustering of PI4,5P2, localization of TORC2 to these clusters and the resulting shutdown of TORC2-Ypk1/2 signaling (Nat Cell Biol 2018, 20(9)). In this context, the shutdown of sphingolipid synthesis makes sense since the plasma membrane experiences low tension (too many lipids). The question is: does edelfosine act similarly? Does edelfosine cause loss of plasma membrane tension? Does PalmC cause activation of the CWI pathway? Loss of plasma membrane tension could explain the increased stabilization of the eisosomes and the increased localization of Slm1/2 to eisosomes.

We thank the reviewer for this constructive suggestion. As the reviewer has pointed out, PalmC reduces a plasma membrane tension and emerges an invaginated plasma membrane structure in which PtdIns(4,5)P₂ accumulates (called PES) (Nat Cell Biol 2018, 20(9)). The formation of PESs is also observed by hyper-osmotic shock that result in a decrease in plasma membrane tension (Nat Cell Biol 2018, 20(9)). Indeed, structurally edelfosine and PalmC are similar; however, no clear formation of PESs was observed following treatment with edelfosine in our experimental conditions as shown in an attached figure below. Additionally, an overexpression of active Pkc1 also did not induce the PES formation. Meanwhile, it is generally understood that the CWI pathway is activated under conditions that increase the plasma membrane tension, such as hypo-osmotic shock, and conversely, hyper-osmotic shock, which reduces the plasma membrane tension and induces PES formation as well as PalmC, is known to depress the activity of CWI pathway (J. Biol. Chem. 270:30157-30161, 1995; J. Biol. Chem. 277:21278-21284, 2002). Therefore, we believe that edelfosine does not cause loss of the plasma membrane tension and inhibits TORC2-Ypk1/2 signaling through a different mechanism from that of PalmC, i.e., through activation of CWI pathway. The mechanism underlying how edelfosine activates the CWI pathway is very interesting, and we are focusing

on the effect of edelfosine on Wsc1 and Mid2 (sensors for CWI pathway), which would be the subject of future research.

We added and modified relevant sentences in the revised manuscript (lines 427-429: “**Additionally, PtdIns(4,5)P₂ has been also reported to be involved in negative regulation of TORC2-Ypk1/2 signaling through a formation of PESs triggered by a decrease in plasma membrane tension (Riggi *et al.*, 2018).**”, lines 431-434: “Additionally, in microscopic analysis of the subcellular distribution of PtdIns(4,5)P₂ using GFP-2xPH^{PLCd} (Stefan *et al.*, 2002), an enrichment of GFP-2xPH^{PLCd} at eisosomes **and the PES formation were** not observed in *PKC1^{R398P}*-expressing cells (data not shown).”).

2) To test for physiological relevance, the authors should include in their studies hyperosmotic stress, which is known to induce the CWI pathway. Again, this might be caused by the loss of plasma membrane tension (cells shrink and PM tension is lost). Does hyperosmotic shock cause the shutdown of TORC2-Ypk1/2 signaling via *Pkc1*?

As mentioned above, the CWI pathway in *S. cerevisiae* is basically considered to be activated when the plasma membrane tension is increased. On the other hand, as the reviewer has pointed out, there is indeed a study showing that the CWI pathway is weakly and transiently activated in response to hyper-osmotic shock through the high-osmolarity glycerol (HOG) pathway that produces the compatible osmolyte glycerol (FEBS Lett. 579:6186-6190, 2005). The activation of the CWI pathway induced by hyper-osmotic shock occurs after a considerable delay, as the activity of HOG pathway returns to basal levels. It has been suggested that the hyper-osmotic shock-induced activation of the CWI pathway may be the result of an overshoot in intracellular glycerol concentrations, which would increase turgor pressure (Fungal Biol. 124:361-367, 2020). Therefore, the loss of plasma membrane tension is not likely to be a direct cause of the CWI pathway activation, and we believe that the inhibitory effect of *Pkc1* on the TORC2-Ypk1/2 signaling is not exerted by loss of the plasma membrane tension.